# When does a Lotka-Volterra model represent microbial interactions? Insights from *in vitro* nasal bacterial communities

Sandra Dedrick,[1] Vaishnavi Warrier,[1] Katherine P. Lemon,[2] Babak Momeni[1]

**ABSTRACT** To alter microbial community composition for therapeutic purposes, an accurate and reliable modeling framework capable of predicting microbial community outcomes is required. Lotka-Volterra (LV) equations have been utilized to describe a breadth of microbial communities, yet, the conditions in which this modeling framework is successful remain unclear. Here, we propose that a set of simple *in vitro* experiments—growing each member in cell-free spent medium obtained from other members—can be used as a test to decide whether an LV model is appropriate for describing microbial interactions of interest. We show that for LV to be a good candidate, the ratio of growth rate to carrying capacity of each isolate when grown in the cell-free spent media of other isolates should remain constant. Using an *in vitro* community of human nasal bacteria as a tractable system, we find that LV can be a good approximation when the environment is low-nutrient (i.e., when growth is limited by the availability of nutrients) and complex (i.e., when multiple resources, rather than a few, determine growth). These findings can help clarify the range of applicability of LV models and reveal when a more complex model may be necessary for predictive modeling of microbial communities.

**IMPORTANCE** Although mathematical modeling can be a powerful tool to draw useful insights in microbial ecology, it is crucial to know when a simplified model adequately represents the interactions of interest. Here, we take advantage of bacterial isolates from the human nasal passages as a tractable model system and conclude that the commonly used Lotka-Volterra model can represent interactions among microbes well when the environment is complex (with many interaction mediators) and low-nutrient. Our work highlights the importance of considering both realism and simplicity when choosing a model to represent microbial interactions.

**KEYWORDS** microbial communities, nasal microbiota, mathematical modeling, community ecology, microbial ecology, microbial interactions

D
ysbiosis in the human-associated microbiota has been associated with several diseases (1, 2). Recently, microbiota-based therapies (e.g., microbiota transplants (3) or metabolite-based therapies (4)) have gained traction as a viable treatment option for diseases associated with dysbiosis (5). Early applications of such therapies have shown great promise but have also exposed important knowledge gaps. It remains unclear what strategies will work best to manipulate the complex, dynamic ecosystem of human microbiota. A comprehensive understanding of interspecific and abiotic interactions is required to understand how diversity is maintained in health, how perturbations lead to dysbiosis in disease, and how best to alter the community composition to maintain or regain health.

Different approaches have been utilized to investigate the structure and function of microbial communities, including human cross-sectional and longitudinal studies (6–11), *in vivo* studies in animal models (12–16), and *in vitro* investigations (17–19). While

Address correspondence to Babak Momeni, momeni@bc.edu.

See the funding table on p. 15.

each approach provides its own level of resolution and insight, each also has its own unique set of limitations. Human studies can offer a broad overview of host-microbe associations but are primarily descriptive in nature. Limited control over *in vivo* systems limits the extent of mechanistic studies in such systems. Conversely, *in vitro* experiments can provide evidence of the specific mechanisms at play but are limited in their ability to emulate *in vivo* conditions. To reconcile these distinct phenomenological and mechanistic inquiries, a compatible mathematical model that can connect the specific mechanisms to observed changes will be a powerful tool.

Previous studies have considered the use of simplified models, including standard, generalized, and compositional Lotka-Volterra (LV) models (20–22), to represent microbial communities. In some cases, this approach has been successful. For instance, it allowed predicting interspecies interactions (23), species coexistence (24), and even community structure and dynamics (17, 18, 25). However, there are concerns about the applicability of such simplified models for representing the community; simpler models may inadequately capture different types of interactions (26, 27), or omit higher-order interactions (26, 28). Thus, the question of when an LV-type model is a good approximation for a community remains open.

In this study, we use *in vitro* communities of human nasal bacteria to investigate when an LV-type model is appropriate for describing microbial community dynamics. For this, we tested nasal bacteria in fresh media and in cell-free spent media (CFSM) of other isolates to quantify their interactions, motivated by the idea that changes in the chemical environment by each species is a major driver of its interactions with its cohabitants (29). Human nasal isolates' growth parameters derived from these *in vitro* experiments were then used to appraise an LV-type model. Our data show that in low-nutrient environments, bacterial growth rate (i.e., how fast bacteria grow) and carrying capacity (i.e., to what extent bacteria grow) obtained in the CFSMs of other isolates are positively and linearly correlated. Under these conditions, only a single parameter which we call "habitat quality" is enough to capture each isolate's influence on another. We show that this linear correlation between growth rate and carrying capacity is consistent with an LV model. Additionally, we conducted coculture experiments and compared experimental results to the model-derived outcomes. Our coculture experiments under low-nutrient conditions confirm that LV-type models can adequately approximate experimental outcomes in cocultures of two nasal isolates.

## MATERIALS AND METHODS

### Bacterial strains

Primary nostril isolates were isolated from the nasal vestibule of two adult volunteers under an Institutional Review Board-approved protocol, as previously described (30): one *Staphylococcus aureus* carrier and one non-carrier. Strains isolated from the *S. aureus* carrier include *S. aureus* KPL1828, *Corynebacterium* sp. KPL1821, and the two non-aureus strains *Staphylococcus* sp. KPL1839 and *Staphylococcus* sp. KPL1850. Strains derived from the non-carrier include the non-aureus strain *Staphylococcus* sp. KPL1867 and *Corynebacterium pseudodiphtheriticum* KPL1989. The fluorescent *S. aureus* (*sGFP S. aureus* Newman) used in cocultures was provided by Dr. Nienke de Jong (Novartis Pharma, Randstad, the Netherlands) (31).

### General experimental setup

Unless otherwise specified, to revive cells from frozen stocks, bacterial isolates were grown in 3 mL of 100% Todd Hewitt Broth + 0.5% Yeast Extract + 1% Tween80 (THY+T80) until mid-exponential growth phase (typically to an $OD_{600}$ between 0.15 and 0.3). Cells from each monoculture were then centrifuged down and re-suspended in 1× phosphate-buffered saline (PBS) before transferring a small fraction into the specific testing condition.

## Cell-free spent medium exposure experiments

All CFSM exposure experiments were carried out in MOPS-buffered THY+T80 diluted to 10% concentration and set to pH 7.2 (10% THY+T80). This medium condition was selected since it moderately reduces the growth rate and the carrying capacity of each isolate, allowing us to quantify both increases (facilitation) and decreases (inhibition) in growth when in the presence of another isolate's CFSM. CFSMs were prepared by inoculating 12 mL of 10% THY+T80 with cells at an initial $OD_{600}$ of 0.01 (~$8 \times 10^6$ CFUs/mL). Cultures were grown for ~16–18 h at 37°C (with shaking), allowing all isolates to reach stationary phase. CFSM was isolated by centrifuging down the cells (4,000 rpm for 10 min) and filtering the supernatant through a 0.22-µm syringe filter. The pH of the CFSM was then re-adjusted (to pH 7.2). For the *Staphylococcus* isolates, a 0.1 $OD_{600}$ stock was prepared from monocultures grown to mid-exponential phase, and 50 µL of this stock was added to 1 mL of each of the isolate's CFSM (including its own), bringing the final $OD_{600}$ to 0.005 for subsequent growth curve experiments. A 0.3 $OD_{600}$ stock was prepared for *Corynebacterium* sp. KPL1821 and *C. pseudodiphtheriticum* KPL1989, and 150 µL of this stock was added to 1 mL of each isolate's CFSM, bringing the $OD_{600}$ to 0.05 for subsequent growth curve experiments. All isolate-CFSM combinations were transferred to a sterile 384-well microplate (Greiner Bio-One, USA) (75 µL × four replicates).

The $OD_{600}$ of each well was read every 10 min for 24–48 h using a BioTek Epoch 2 microplate reader (www.biotek.com, now Agilent). Growth rate and carrying capacity (using maximum $OD_{600}$ as a proxy) were calculated using code developed in MATLAB R2018. See Fig. 1 for experimental setup. Carrying capacities are estimated based on the maximum $OD_{600}$ values reached for monocultures within 48 h of growth. Growth rates are calculated by fitting a linear line into the log-transformed OD readings in early stages of growth (typically in the $OD_{600}$ range below 30% of the carrying capacity). Fig. S1 shows representative growth curves obtained in these experiments.

## Growth experiments

All nasal isolates were grown in 100% THY+T80 and prepared as described under the section General experimental setup. To identify growth characteristics of nasal isolates in different environmental conditions, we grew each strain in various types of media ranging in concentration from 100% to 0.32%; in the presence of metabolic byproducts and antibiotics; and at various environmental pH values. Conditions used are listed in Table 1. See Fig. 1 for experimental setup.

## Composition of cultivation media

THY medium was made using the premixed powders of Todd Hewitt broth (BD Bacto Dehydrated Culture Media: Todd Hewitt broth, Thermo Fisher Scientific, DF0492-17-6) and yeast extract (Thermo Scientific Yeast Extract, Ultrapure, AAJ23547A1). BHI medium was made using the premixed powder (BD Bacto Dehydrated Culture Media: Brain Heart Infusion, 237500). For both THY and BHI, powders were mixed into tap water and then autoclaved for sterility.

Baseline Defined Media with amino acids (BAAD) medium consists of 1.5 g/L of $KH_2PO_4$, 3.8 g/L of $K_2HPO_4$ ($\times 3H_2O$), 1.3 g/L of $(NH_4)_2SO_4$, 10 g/L of MOPS, 3 g/L of sodium citrate ($\times 2H_2O$), 10 mL/L of the mixed vitamin stock (2 mg/L of biotin, 2 mg/L of folic acid, 10 mg/L of pyridoxine-HCl, 5 mg/L of thiamine-HCl $\times 2H_2O$, 5 mg/L of riboflavin, 5 mg/L of nicotinic acid, 5 mg/L of D-Ca-pantothenate, 0.1 mg/L of vitamin B12, 5 mg/L of p-aminobenzoic acid, and 5 mg/L of lipoic acid), 1 mL of SL-10 mixed trace elements stock (10 mL/L of HCl (25%; 7.7 M), 1.5 g/L of $FeCl_2$ $\times 4H_2O$, 70 mg/L of $ZnCl_2$, 0.1 g/L of $MnCl_2$ $\times 4H_2O$, 6 mg/L of $H_3BO_3$, 0.19 g/L of $CoCl_2$ $\times 6H_2O$, 2 mg/L of $CuCl_2$ $\times 2H_2O$, 24 mg/L of $NiCl_2$ $\times 6H_2O$, and 36 mg/L of $Na_2MoO_4$ $\times 2H_2O$), 5 mL/L of 1M $MgCl_2$, 1 mL/L of 1M $CaCl_2$, 100 mL/L of the mixed amino acids and nucleic acids stock (1.6 g/L of alanine, 1 g/L of arginine, 0.4 g/L of asparagine, 2 g/L of aspartic acid, 0.05 g/L of cysteine, 6 g/L of glutamic acid, 0.12 g/L of glutamine, 0.8 g/L of glycine, 1 g/L of histidine

**TABLE 1** Media and environmental conditions used to characterize the growth of nasal isolates

| Media | Environmental condition | Concentrations (% or µg/mL) |
|---|---|---|
| Todd Hewitt broth (THY+T80) pH 7.2 | N/A[d] | 100%, 50%, 20%, 10%, 5%, 2.5%, 1.25%, 0.63%, or 0.32% |
| Brain heart infusion (BHI+T80) pH 7.2 | N/A | 100%, 50%, 20%, 10%, 5%, 2.5%, 1.25%, 0.63%, or 0.32% |
| Baseline media + amino acids (+T80) | N/A | 100%, 75%, 50%, 25%, 15%, 10%, 5%, 2.5%, and 1.25% |
| 10% THY+T80 (MOPS-buffered) pH 7.2 | Metabolic byproduct, acetic acid[a] | 0%, 0.2%, 0.4%, 0.6%, 0.8%, and 1.0% |
| 10% THY+T80 (MOPS-buffered) pH 7.2 | Metabolic byproduct, lactic acid[b] | 0%, 0.2%, 0.4%, 0.6%, 0.8%, and 1.0% |
| 10% THY+T80 (MOPS-buffered) pH 7.2 | Antibiotic, vancomycin[c] | 0, 0.025, 0.050, 0.100, 0.125, 0.150, 0.175, 0.200, or 0.250 µg/mL |
| THY+T80 (MOPS-buffered) | Various environmental pH | pH 5.1, 5.4, 5.7, 6.0, 6.3, 6.6, 6.9, 7.2, 7.5 |

[a] Glacial acetic acid, Fisher Chemical, Fisher Scientific.
[b] L(+)-lactic acid (90% solution in water), ACROS Organics, Fisher Scientific.
[c] Vancomycin hydrochloride, Millipore Sigma.
[d] N/A, not applicable.

monohydrochloride monohydrate, 2 g/L of isoleucine, 2.6 g/L of leucine, 2.4 g/L of lysine monohydrochloride, 0.6 g/L of methionine, 2 g/L of phenylalanine, 2 g/L of proline, 1 g/L of serine, 0.7 g/L of threonine, 0.3 g/L of tryptophan, 0.25 g/L of tyrosine, 2 g/L of valine, 2 g/L of adenine hemisulfate salt, and 2 g/L of uracil), 1.10 mg/L of $FeSO_4 \times 7H_2O$, and 0.4% final dextrose.

## Derivation of a linear relationship between growth rate and carrying capacity of CFSMs based on an LV formulation of interactions

Assume the standard logistic growth for each of the isolate:

$$\frac{dS}{dt} = r\left(1 - \frac{S}{K}\right)S$$

Assuming an LV model, the presence of another species modulates the growth rate proportionally to the size of the interacting partner, that is,

$$\frac{dS_1}{dt} = r_1\left(1 - \frac{S_1 + c_{12}S_2}{K_1}\right)S_1$$

Calculating the parameters obtained from the CFSM, growth of isolate 1 in the CFSM of isolate 2 can be represented as

$$\frac{dS_1}{dt} = r_1\left(1 - \frac{S_1 + c_{12}K_2}{K_1}\right)S_1$$

since in the CFSM of isolate 2, the environment will resemble the situation at which the density of $S_2$ has reached $K_2$. We estimate the growth rates when $S_1$ is very small, which means when

$$\frac{dS_1}{dt} \approx r_1\left(1 - \frac{c_{12}K_2}{K_1}\right)S_1$$

Therefore, growth rate in the CFSM assay is

$$r_{12} = r_1\left(1 - \frac{c_{12}K_2}{K_1}\right)$$

The carrying capacity for isolate 1 is reached at population $S_1$ level when growth rate becomes 0, thus

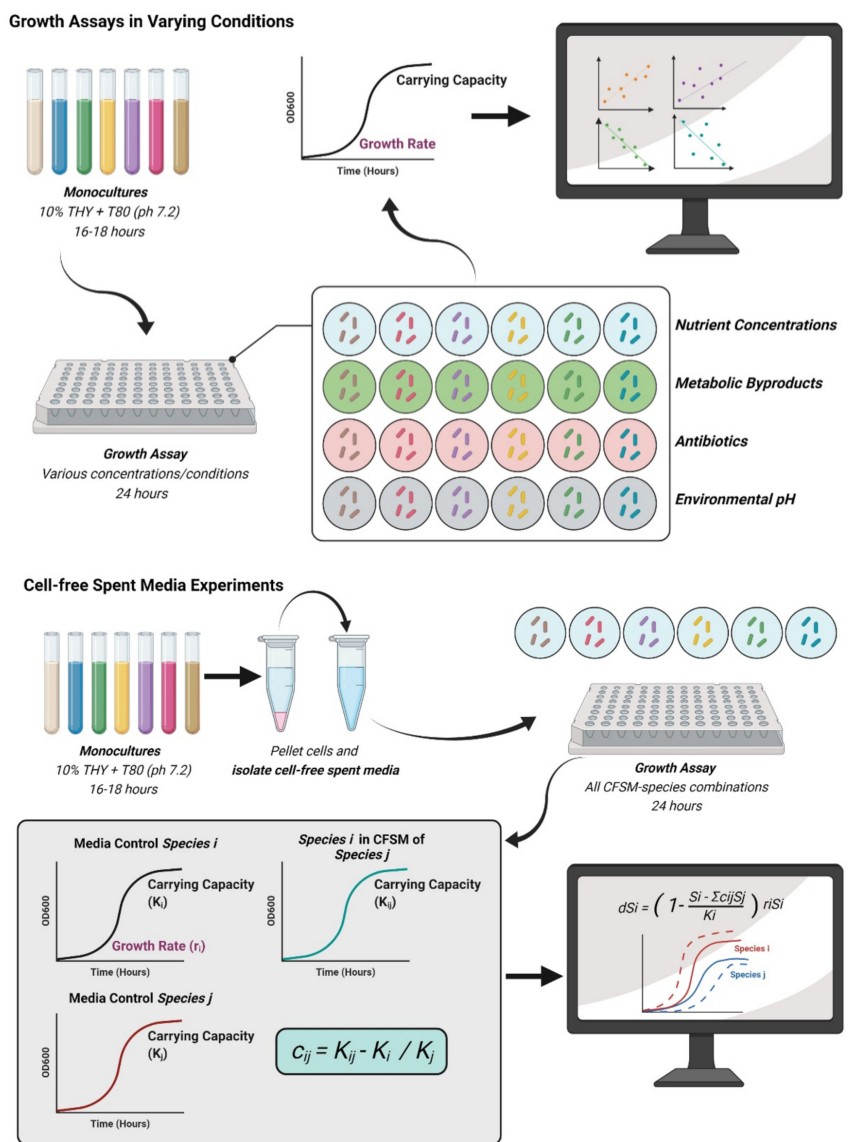

**FIG 1** Experimental setup for growth assays. (A) Experimental setup for growth assays in varying environmental conditions. Nasal isolates were grown for 16–18 h in 12 mL of 10% THY+T80 (pH 7.2) and then transferred into different environmental conditions (e.g., various nutrients concentrations, metabolic byproduct concentrations, antibiotic concentrations, and environmental pH). For each isolate-condition combination, a linear regression was performed on growth rate and carrying capacity data to determine the relationship between these two growth characteristics. (B) Experimental setup for cell-free spent media (CFSM) conditions. Nasal isolates were grown for 16–18 h in 12 mL of 10% THY+T80 (pH 7.2) and supernatants were filter-sterilized. Monocultures of each isolate were also grown under the same conditions. All isolates were then grown in the presence of all other CFSM monocultures. Data derived from CFSM experiments were used to inform an LV-type model. Correlation coefficients were calculated using the carrying capacity from each isolate grown in standard medium (baseline growth characteristics), along with the carrying capacity of each isolate grown in the presence of one another's CFSM. Simulated outcomes were compared to the experimental outcomes from coculture experiments. LV, Lotka-Volterra. Fig. 1 was created by BioRender.com.

$$\left(1 - \frac{S_{1,cc} + c_{12}K_2}{K_1}\right) = 0$$

Therefore, the carrying capacity in the CFSM assay is

$$K_{12} = S_{1,cc} = K_1 - c_{12}K_2$$

Calculating the ratio of growth rate to carrying capacity, we obtain

$$\frac{r_{12}}{K_{12}} = \frac{r_1(1 - \alpha_{12}K_2/K_1)}{K_1 - \alpha_{12}K_2} = \frac{r_1(K_1 - \alpha_{12}K_2)}{K_1(K_1 - \alpha_{12}K_2)} = \frac{r_1}{K_1}$$

In other words, the ratio of growth rate to carrying capacity in CFSM assays is independent of the isolate used for the CFSM. This $r/K$ ratio is the slope of the line found in the empirical data.

## Two-species cocultures to validate the model

Coculture enrichment experiments were performed in 10% THY+T80 media buffered with 10 g/L of MOPS. Cocultures were performed for *sGFP S. aureus* Newman-*Corynebacterium* sp. KPL1821 and *sGFP S. aureus* Newman-*Staphylococcus* sp. KPL1850. All isolates were grown in 100% THY+T80 and prepared as described under the section "General experimental setup." Species were then mixed in a 1:1, 1:10, and 1:100 ratio (*S. aureus*: Species 2) at a 0.01 starting OD$_{600}$ and distributed into a sterile 96-well microplate (×6 replicates each). Plates were read in a BioTek Synergy Mx microplate reader (www.biotek.com) set to 37°C with continuous shaking. The OD$_{600}$ and fluorescence (485 nm excitation and 528 nm emission) of each well was read every 10 min for 24 h. Serial dilutions of each replicate were plated on 100% THY+T80 agar medium to verify plate reader results.

## Comparison of cocultures with an LV model derived from CFSM experiments

To calculate the OD$_{600}$ for *Staphylococcus* sp. KPL1850, GFP signal from *sGFP S. aureus* was converted into OD$_{600}$ using a calibration curve (GFP vs. OD$_{600}$ derived from *sGFP S. aureus* monoculture) based on exponential growth (Fig. S2).

$$\text{OD}_{600} = \begin{cases} 2.97 \times 10^{-5} \cdot (F + 200) & ; if \ F < 3000 \\ -a + \sqrt{a^2 - b} & ; if \ F \geq 3000 \end{cases},$$
$$\text{with a} = 0.0283 \text{ and b} = 5.20 \times 10^{-6} \cdot (100 - F)$$

The converted OD$_{600}$ for *S. aureus* OD$_{600}$ was then subtracted from the total OD$_{600}$ to obtain the OD$_{600}$ for the partner strain. At OD values less than 0.5, we found a linear relationship between the OD and the cell densities. At higher OD values, the relationship was no longer linear, but we can still use a conversion to estimate the cell density from the measured OD$_{600}$ (Fig. S3). All analyses and algorithms were created and run in MATLAB R2018 and are shared on GitHub (https://github.com/bmomeni/nasal-community-modeling). Coculture dynamics were simulated based on standard growth parameters for each isolate (growth rate and carrying capacity of monoculture in 10% THY+T80) and the carrying capacity of each isolate grown in the presence of other isolates' CFSM. When modeling the growth of monocultures, or cocultures, in the absence of interactions, we assume that growth follows standard logistic growth

$$\frac{dS_i}{dt} = \left(1 - \frac{S_i}{K_i}\right)r_iS_i$$

in which the population density at a given time ($dS_i/dt$) is determined based on the isolates' baseline growth rate ($r_i$), carrying capacity ($K_i$), and the current population

density ($S_i$). Here, the growth rate of an isolate is modulated by the fraction of the maximum population for isolate $i$ $(1 - S_i/K_i)$, so as the population density nears its maximum carrying capacity, the growth rate decreases modularly.

When interactions are present in a coculture, we assume that the presence of isolate $j$ modulates growth of isolate $i$ $(i, j = 1, ..., N)$.

$$\frac{dS_i}{dt} = \left(1 - \frac{S_i - \sum_{j=1}^{N} c_{ij} S_j}{K_i}\right) r_i S_i$$

$$c_{ij} = \frac{K_{ij} - K_i}{K_j}$$

Similar to logistic growth, the growth rate of each isolate depends on its population size and carrying capacity. However, the "effective" growth rate is modulated by the presence of other isolates in the environment. $K_i$ and $r_i$ represent the carrying capacity and growth rate of isolate $i$ in a monoculture, respectively. $K_{ij}$ represents the carrying capacity of isolate $i$ in the CFSM of isolate $j$, capturing the effect of isolate $j$ on isolate $i$. This formulation is consistent with an LV model in which the coefficients of interactions are $a_{ij} = c_{ij} r_i/K_i$.

When comparing the coculture experimental data to the modeling results, additional lag times (typically around 2 h) were included to account for the delayed growth observed empirically. All experimental replicates were plotted for each experimental condition along with modeling outcomes (Fig. 5). The interaction coefficient was then calculated for the empirical data over the course of exponential growth (first 4–5 h of growth).

## Model parameters

Unless otherwise noted, the parameters listed in Table 2 were included in the model.

## RESULTS

### Nasal microbiota as a model for microbiota-based therapies

The nasal passage is a reservoir for diverse taxa and is a first-line defense against pathogenic invasion and subsequent infection (32–34). It is also home to pathobionts, such as *S. aureus*, which can cause life-threatening infection (35). Previous work has demonstrated that nasal microbiota can be altered using microbiota-based therapy (36, 37). In one such study, the direct nasal administration of the nasal *Corynebacterium* sp. Co304 resulted in the eradication of *S. aureus,* making this microbiome a

**TABLE 2** Model parameters used to simulate coculture results (isolate 1 is *S. aureus* and isolate 2 is non-aureus *Staphylococcus* sp. KPL1850)

| Parameter | Description | Value |
|---|---|---|
| $R_1$ | Growth rate (1/h) of isolate 1 | 1.85 |
| $R_2$ | Growth rate (1/h) of isolate 2 | 1.56 |
| $K_1$ | Maximum carrying capacity (OD$_{600}$) for isolate 1 | 0.61 |
| $K_2$ | Maximum carrying capacity (OD$_{600}$) for isolate 2 | 0.35 |
| $K_{12}$ | Maximum carrying capacity (OD$_{600}$) for isolate 1 in the presence of the CFSM of isolate 2 | 0.17 |
| $K_{21}$ | Maximum carrying capacity (OD$_{600}$) for isolate 2 in the presence of the CFSM of isolate 1 | 0.03 |
| $c_{12}$ | Interaction coefficient for isolate 1 (effect of isolate 2 on isolate 1 growth) | −1.26[a] |
| $c_{21}$ | Interaction coefficient for isolate 2 (effect of isolate 1 on isolate 2 growth) | −0.52[a] |
| $S_{10}$ | Initial OD$_{600}$ of isolate 1 | 0.0005–0.005[b] |
| $S_{20}$ | Initial OD$_{600}$ of isolate 2 | 0.005 |
| $T_{1L}$ | Lag time for isolate 1 (h) | 2.35 |
| $T_{2L}$ | Lag time for isolate 2 (h) | 2.25 |

[a] The interaction coefficient ($c_{ij} = (K_{ij} - K_i)/K_j$) estimated using CFSM characterizations in 10% THY.
[b] This varies depending on the starting ratio (1:1 or 1:10).

likely candidate for future microbiota-based therapeutics (37). Former research has also demonstrated the importance of interbacterial interactions in shaping the composition of this community, and a number of microbiont-produced compounds that inhibit the colonization of pathogens, such as *S. aureus*, have been identified (38–42). These attributes enable us to study interactions *in vitro* and provide a useful platform for developing a mathematical model. We also chose to use nasal microbiota as our model consortia since (i) it is relatively low in diversity, numerically dominated by 3-8 species (43), (ii) common nasal colonizers are culturable and able to be studied using *in vitro* laboratory experiments, and (iii) the nasal passage can be easily sampled. See Materials and Methods for a list of primary nasal bacterial isolates used in this study.

## Growth rate and carrying capacity are positively correlated in spent media of other nasal isolates

CFSM experiments are commonly used as biostimulants and biocontrol agents to investigate bacterial interactions with other bacteria, viruses, and eukaryotes (44–49). The factors assayed in these experiments can include resource consumption, the secretion of metabolic byproducts, the production and secretion of additional nutrients, and/or the production of growth inhibitors (e.g., antibiotics, small molecules). To quantify the total growth effect of individual isolates on one another, we exposed each isolate to the CFSM derived from monocultures (see Materials and Methods). A linear regression model revealed a positive relationship between the growth rate and carrying capacity of each isolate grown in the CFSMs of our nasal isolates, including its own (linear regression $p$-value < 0.05 in all cases). This suggests that CFSM modulates growth rate and the carrying capacity of nasal isolates in a similar manner (Fig. 2).

We note that a major motivation for assessing microbial interactions using CFSM experiments is the feasibility of performing these experiments in different contexts,

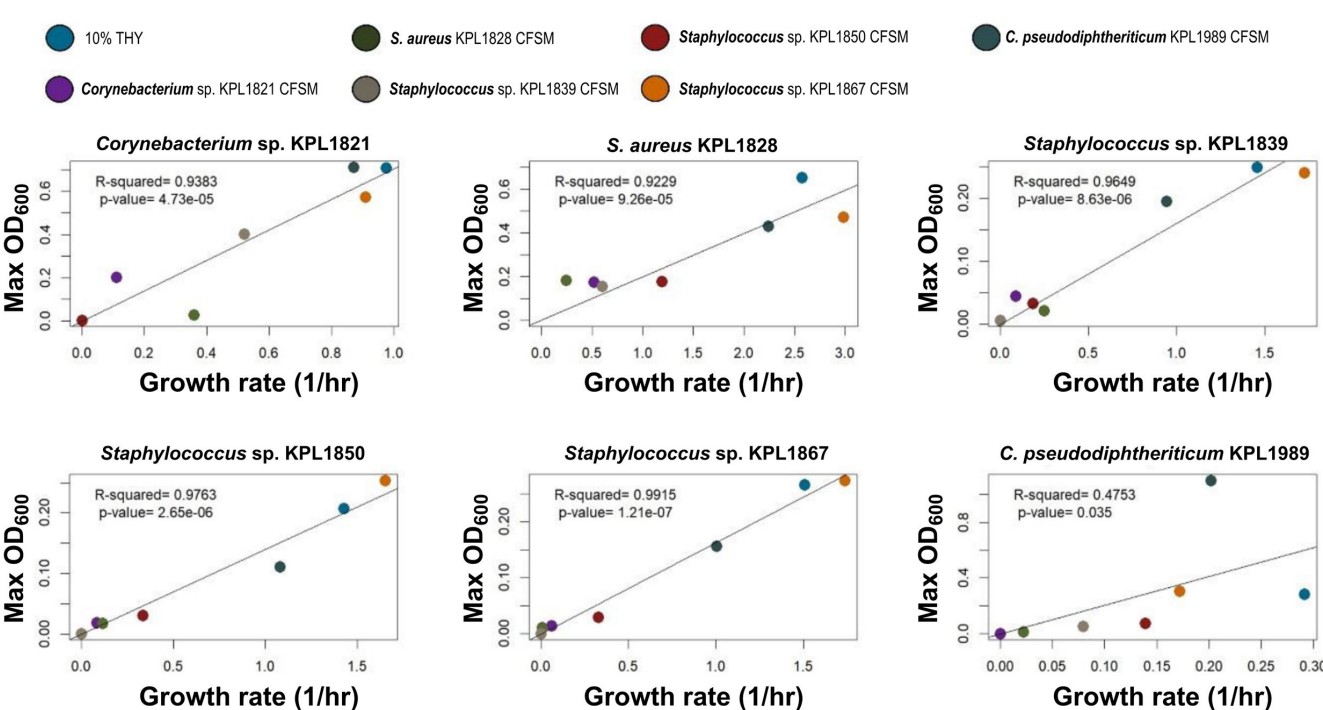

**FIG 2** Growth rate-carrying capacity results from nasal isolates grown in other nasal isolate's CFSM show a consistent positive correlation. Colors of data points represent the context (10% THY for CFSM from one of the strains). Each panel shows the growth of one of the strains (as labeled on top) in different contexts. A linear regression analysis (solid line and corresponding $R^2$) reveals a strong positive relationship between growth rate and carrying capacity. Each data point shows the average growth rate and carrying capacity (using maximum measured $OD_{600}$ as a proxy for the carrying capacity) from two independent experiments with 3-6 replicates per experiment.

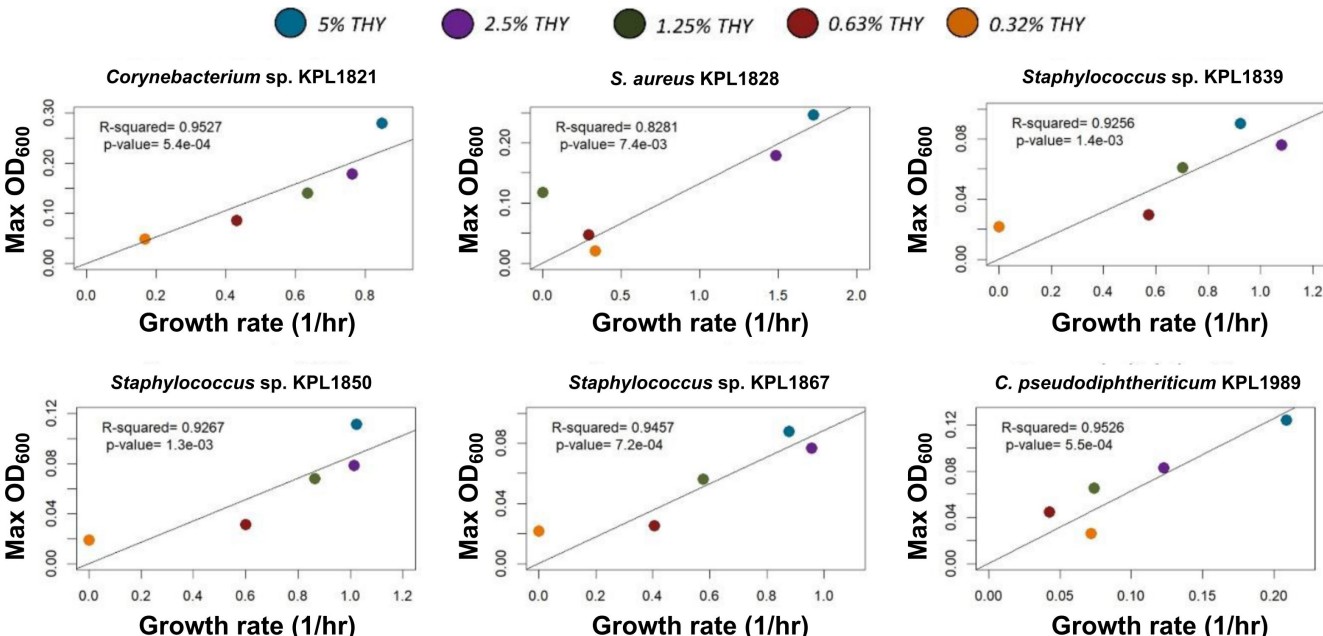

**FIG 3** Growth rate-carrying capacity results from nasal isolates grown in low concentrations (0.32%–5%) of THY show a consistent positive correlation. Colors of data points represent the context (different dilutions of THY). Each panel shows the growth of one of the strains (as labeled on top) in different contexts. A linear regression analysis (solid line and corresponding $R^2$) reveals a strong positive relationship between growth rate and carrying capacity when isolates are grown in low-nutrient concentrations. Each data point shows the average growth rate and carrying capacity (using maximum measued $OD_{600}$ as a proxy for carrying capacity) from two independent experiments with 3–6 replicates per experiment.

only requiring the ability to measure growth properties in monocultures. However, these experiments will not comprehensively capture all possible interaction mechanisms. Notable interactions, such as those that rely on physical/mechanical contact between cells or those that are triggered only when a partner is present, will not be captured in basic CFSM experiments. Nevertheless, because many important interactions are represented in CFSM experiments, this approach is routinely used to assess microbial interactions (17, 19, 50–52).

## Correlated growth rate and carrying capacity are consistent with an LV model of bacterial interactions

We can show that the positive, linear correlation observed in our data is consistent with an LV representation of interactions between bacteria (see Materials and Methods). Assuming an LV model, the growth rate ($r_{ij}$) and carrying capacity ($K_{ij}$) of isolate *i* in the CFSM of isolate *j* are found to be

$$\frac{r_{ij}}{K_{ij}} = \frac{r_i}{K_i}$$

which depends on the growth rate and carrying capacity of the nasal isolate being tested ($r_i \ and \ K_j$), and not the properties of the bacterium used for the spent medium. Thus, we hypothesize that the growth rate-carrying capacity relationship derived from CFSM experiments can be used to indicate whether an LV-like model is a suitable framework for describing pairwise strain interactions.

## Growth rate-carrying capacity correlations are consistently observed in complex, low-nutrient environments

In our data, we observe a positive association between growth rate and carrying capacity in CFSM experiments. In fact, for ~50% of CFSM conditions, bacteria grow to <30% of their carrying capacity potential and ~80% of the low carrying capacity interactions also have a low growth rate. One explanation is that CFSM is depleted of nutrients required for growth. To test this, we grew each bacterial isolate in a range of nutrient concentrations to determine if growth rate and carrying capacity are similarly modulated by environmental concentrations of growth-limiting nutrients (i.e., carbon, nitrogen, and phosphorus). As described in the Materials and Methods section, bacteria were grown in different concentrations (100%–0.32%) of two different rich media, THY+T80 and BHI+T80. Both media contain a range of nutrients required for bacterial growth at high concentrations, making them both complex and rich in nature. In these experiments, we observed a positive correlation between growth rate and carrying capacity at low-nutrient concentrations (≤5%) (Fig. 3 and Fig. S4). However, at higher nutrient concentrations (≥10%), the relationship between growth rate and carrying capacity deviates from the positive correlation (Fig. S5). Some isolates, in fact, show a local negative correlation between growth rate and carrying capacity at higher nutrient concentrations, including *S. aureus* KPL1828. These data suggest that the positive relationship between growth rate and carrying capacity only applies to environments with lower concentrations of nutrients. Most microenvironments that support microbial communities are nutritionally complex yet low in concentration (53). Similarly, previous studies on the nasal environment have shown that the nasal passages contain a range of nutrients at low concentrations (54). Thus, an interaction-based model that includes simple growth parameters remains applicable to biologically relevant environments, such as the nasal microbiota.

Previous work in *Escherichia coli* has shown that nutrients, such as carbon and nitrogen, lower bacterial carrying capacity but not growth rate when grown in limiting concentrations (55). To determine if single nutrient sources similarly affect carrying capacity (but not growth rate) for nasal microbiota, we performed similar experiments in a chemically defined medium, BAAD. Here, we grew each isolate in a range of media dilutions between 100% and 1.25%. When we compare carbon and nitrogen concentrations to each isolate's growth rate and carrying capacity separately, we observe a strong linear and positive correlation between carbon concentrations and carrying capacity, but not growth rate (Fig. S6). These data indicate a stronger correlation between the nutrient concentration with the carrying capacity compared to the growth rate.

## Modulating an individual growth mediator may not lead to the positive linear trend in the growth rate-carrying capacity relation

Bacteria can influence other bacteria by changing their local microenvironment. This can take place through a variety of compounds. We picked three representative examples to explore how modulation of the microenvironment might affect bacterial interactions. We examined interactions through the production of metabolic intermediates, antibiotic compounds, and alteration of the environmental pH. To determine if the relationship between growth rate and carrying capacity is conserved in the presence of such factors commonly found in the bacterial microenvironment, we performed monoculture experiments in increasing concentrations of two common bacterial metabolic byproducts, acetic acid and lactic acid; in increasing concentrations of vancomycin; and in the presence of a range of pH values.

In these conditions, we observed a variety of possible relationships between growth rate and carrying capacity for most isolates in the presence of metabolic byproducts, antibiotics, and in a range of different environmental pH values (Fig. 4 and Fig. S7, S8). Interestingly, the association between growth rate and carrying capacity was negatively correlated for non-aureus *Staphylococcus* strains in the presence of both acetic and lactic acid. This effect is due to the increased carrying capacity in the presence of increasing

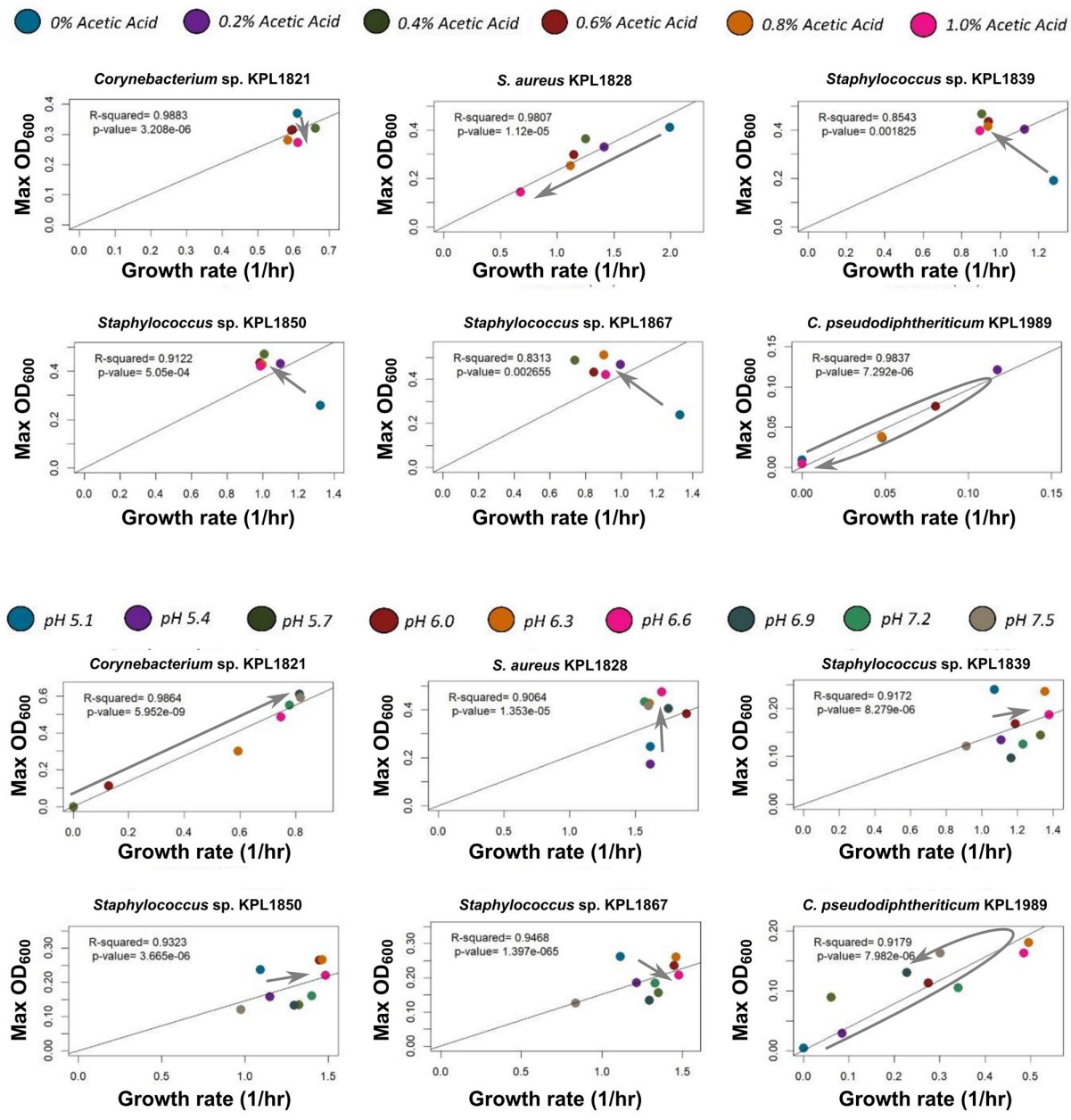

**FIG 4** Growth rate-carrying capacity results from nasal isolates grown after modulating various mediators are isolate- and mediator-specific. In 10% THY+T80 as the base medium, we tested different concentrations of acetic acid (from 0% to 1%; pH adjusted to 7.2) and different pH values from 5.1 to 7.5. Colors of data points represent the context (different acetic acid concentrations supplied on top; different pH values at the bottom). Each panel shows the growth of one of the strains (as labeled on top) in different contexts. The gray arrow is a visual guide to highlight the trend of changes. A range of different trends are observed: insensitivity (e.g., *Corynebacterium* sp. KPL1821's response to acetic acid), inhibition (e.g., *Staphylococcus aureus*' response to acetic acid), best growth at intermediate levels (e.g., *C. pseudodiphtheriticum*'s response to pH and acetic acid), and trading carrying capacity for growth rate (e.g., non-aureus *Staphylococcus*' response to acetic acid). Each data point shows the average growth rate and carrying capacity (using Max OD$_{600}$ as a proxy) from two independent experiments with 3–6 replicates per experiment.

concentrations of either acid, suggesting that the non-aureus *Staphylococcus* strains can utilize these metabolic byproducts.

These data demonstrate that the effect of individual growth mediators on growth rate and carrying capacity are both condition- and strain-specific. However, our CFSM results suggest that a complex environment consisting of a range of nutrients, byproducts, and antimicrobial compounds, affects growth dynamics in a linear and predictable manner.

Thus, despite our inability to apply general principles of growth at the level of individual factors, these principles can be applied to complex environments.

## Coculture dynamics are qualitatively consistent with the model obtained from CFSMs in low-nutrient environments

We used two species cocultures to validate our model. For this, *sGFP S. aureus* Newman and non-aureus *Staphylococcus* sp. KPL1850 cocultures with initial populations of *S. aureus:Staphylococcus* sp. KPL1850=1:1, 1:10, and 1:100 were grown for 24 h (37°C with shaking). When we compared our modeling results to the coculture experimental results, we observed a significant deviation in outcomes. The most distinct difference between the modeling and experimental results was the lag time. After incorporating parameters for lag time (2.35 and 2.25 h for *S. aureus* and non-aureus *Staphylococcus* sp. KPL1850, respectively), modeling results had a significantly improved fit throughout growth (Fig. 5 top).

The interaction coefficient for each isolate was calculated over the course of exponential growth and compared to the model-based interaction coefficient. To do this, the interaction coefficient was calculated after smoothing the experimental data and it was compared with the coefficients derived from the CFSM characterizations.

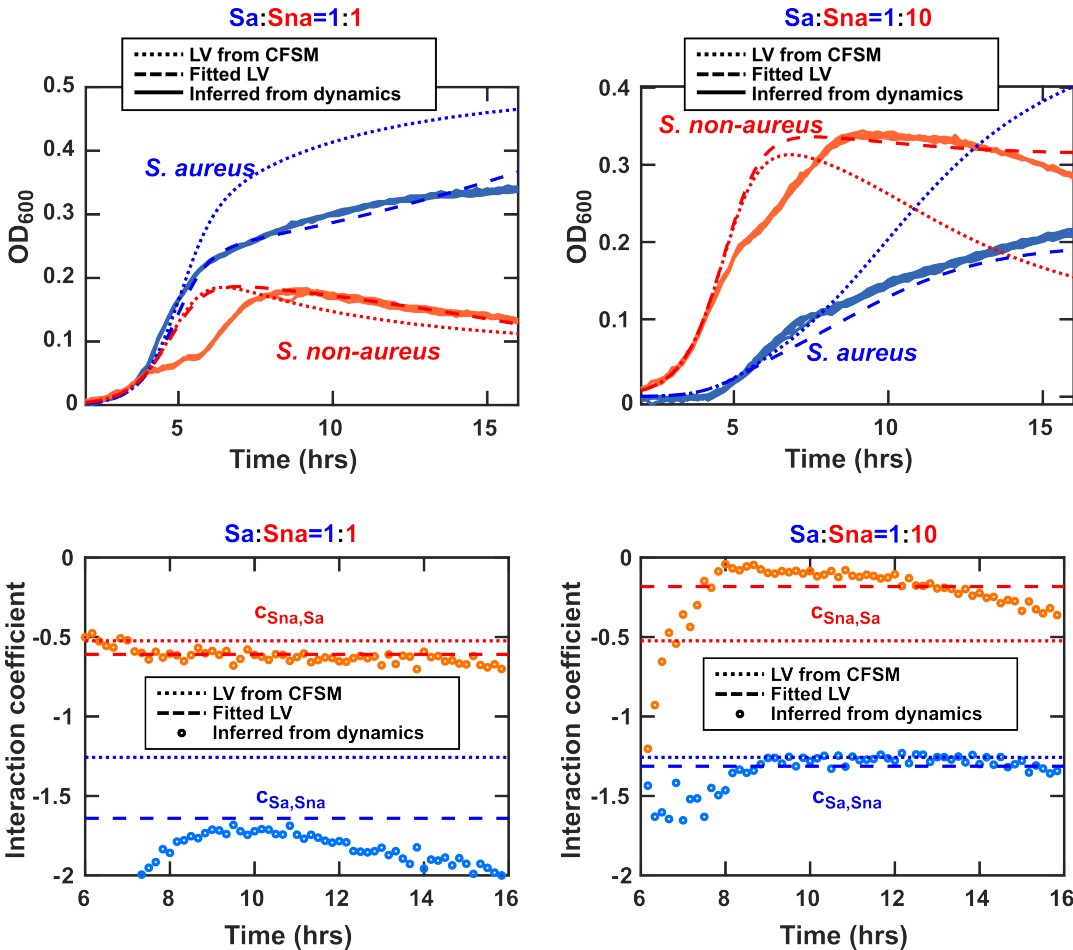

**FIG 5** Comparison of experimental and modeling results for *sGFP Staphylococcus aureus* (Sa) and non-aureus *Staphylococcus* sp. KPL1850 (Sna) cocultures shows that LV models can offer a reasonable approximation. Top: Coculture experiments and simulations were performed as described in the Materials and Methods section. After adjusting for lag time, modeling results were comparable to the experimental outcomes throughout exponential and stationary growth phases. Bottom: Interaction coefficients derived from CFSM characterizations (dotted) approximated the values inferred from measured population dynamics (circles). Fitting an LV model into measured dynamics using the two interaction coefficients as degrees of freedom (dashed) also offered an acceptable approximation. Six technical replicates are used in each case.

The experimental interaction coefficient agreed with the model interaction coefficient in sign (i.e., negative interaction) and remained generally aligned with the model (Fig. 5, bottom).

As a control, we also measured the interaction coefficients from the CFSM characterizations in a rich medium (100% THY). We could still find an LV model that matched the measured dynamics (Fig. S9, top); however, the interaction coefficients for this fitted LV did not match those obtained from CFSM characterizations neither in sign nor in magnitude (Fig. S9, bottom). Importantly, the interaction coefficients highly depended on the initial conditions (Fig. S9), suggesting that a reliable LV model in such rich media does not exist.

To further validate our model, we tested how each nasal isolate performed in 1:1 combined CFSMs of two other isolates. For the model to extend beyond pairwise cocultures, we expect the combined effects of different isolates through the environment to add up linearly. Our results show that, in most cases, this expectation holds up. Although these results need further investigations, in the majority of cases tested, the response to 1:1 combined CFSMs was indistinguishable from the average of responses to each of the CFSMs (Fig. S10).

## DISCUSSION

The LV model is a set of equations that can model trophic relationships between species. The advantage of this model lies in its simplicity. For two interacting species, only six parameters, all easily estimated experimentally, are required: growth rates, carrying capacities, and interspecies interaction coefficients for each. Originally developed to describe competitive and predator-prey interactions, LV models have since been adapted for a broader range of ecological problems. LV equations have been used to create microbial interaction networks to predict community outcomes given a set of defined interactions in both theoretical and empirical settings (56–60). Lake ecosystems (61), the human gut (62, 63) and cheese-associated microbial communities (60, 62) are just a few examples for which simple LV equations could describe microbial community dynamics. Despite the popularity of LV models, their range of applicability has remained obscure, with some of their limitations highlighted before (26–28).

To investigate the range of applicability of LV models, we used *in vitro* communities of human nasal bacteria as a system because of its tractability. In particular, we relied on experimental measurements of growth rate and carrying capacity of each bacterium in CFSM of other strains/species. If an LV representation of interactions is valid, we would expect the growth rate-carrying capacity ratio of a focal isolate grown in the CFSM of a partner isolate to be equivalent to that of the focal isolate grown in a monoculture. Indeed, in low-nutrient complex environments we observed this pattern. This pattern was consistent with the resource availability in low-nutrient environments but did not hold in nutrient-rich conditions. Moreover, when we modulated the concentrations of individual factors commonly found to mediate community structure (such as metabolic byproducts, antibiotics, or pH), results were highly condition- and species/strain-dependent. For metabolic byproducts, the growth rate-carrying capacity relationship was only weakly linear for some nasal isolates, while others displayed a negative correlation. Based on our results, we posit that an LV model can describe community dynamics in complex, low-nutrient environments, despite possible inconsistent effects of individual mediators. We also propose the use of simple CFSM experiments as a criterion to decide whether an LV model is suitable for describing a microbial community of interest. If growth rate and carrying capacity are not linearly correlated for isolates grown in CFSM obtained from other community members, then an LV model is inappropriate for such a system.

We should emphasize that observing a linear relationship between the growth rate and carrying capacity is not sufficient to establish that an LV model is appropriate for representing the community of such interactions—other models may produce the same relationship or higher-order interactions may make the multispecies LV inaccurate.

Instead, we argue that if the linear relationship is not satisfied, LV will not be the right choice for representing the system.

As pointed out by Lipson in his 2015 work, previous research on the correlation between growth rate and carrying capacity has been inconclusive (64). In 1965, Pirt proposed the maintenance energy theory which stated that the amount of energy required for cell maintenance remains the same, despite slow growth rate (65). In this regime, furnishing a more suitable growth condition, for example, by supplying more nutrients, results in an increase in both growth rate and carrying capacity and thus a positive linear relationship between them (65). In contrast, a distinct body of work supports a negative relationship between growth rate and carrying capacity which argues that rate-yield tradeoffs are central to the coexistence of species (66–69). Considering this explicit discrepancy, Lipson reconciled these distinct observations under a single broad perspective (69). He rationalizes that positive growth rate-yield correlations are often observed in low-nutrient or physiologically stressful environments, whereas rate-yield tradeoffs occur under nutrient-rich conditions. Indeed, when nasal isolates were grown in dilutions of a growth medium, we observed a positive growth rate-carrying capacity relationship at lower nutrient concentrations but not at higher nutrient concentrations. Our interpretation, consistent with Lipson's explanation, is that in low-nutrient environments, cells dedicate all of their resources to producing biomass, leading to a strong positive correlation between growth rate and carrying capacity. In nutrient-rich environments, growth rate no longer increases linearly with the nutrient availability and as a result the growth-rate carrying capacity trend deviates from a simple linear relationship. There is also additional evidence that in nutrient-rich environments populations exhibit stronger inhibitory effects (70, 71), which can lead to a decrease in growth rates in cocultures compared to monocultures and thus more deviation from LV equations. Based on our simple derivations of the linear relationship between growth rate and carrying capacity in the section Materials and Methods; thus, only the low-nutrient environments show trends consistent with LV equations. Our CFSM and coculture data using nasal bacteria further support this view.

Even though we posit that the LV model offers an acceptable approximation for the coculture population dynamics (Fig. 5), our results also reveal details in the dynamics that the LV model misses. For instance, there is a slow-down in the growth of non-aureus *Staphylococcus* around hour 6—presumably due to a shift in its metabolism—that the LV model does not capture. Additionally, the LV model derived from CFSM data systematically predicts a higher final *S. aureus* density compared to the results obtained from experiments. These aspects point to more salient aspects of bacterial physiology and interactions, and offer starting points for further investigations.

Some evidence suggests that LV pairwise models fail in communities with higher richness (26, 27, 72), largely due to higher order effects. LV models assume that the fitness of an individual is equal to its basal fitness plus the fitness influences from pairwise interactions ("additivity assumption") (26). This means that in a more complex community of three or more isolates, all fitness influences on an individual are summed up into a single fitness parameter and higher order effects are essentially ignored. Yet, others have shown that pairwise interaction data are sufficient for describing more complex microbial communities, despite the presence of higher order effects (17, 18, 24). In one such study on gut microbiota, Venturelli et al. found that pairwise interaction data were able to adequately predict even multispecies population dynamics (18), supporting the use of simple bottom-up approaches to predict microbial community structure. In this study, we propose a simple, easily testable, framework to determine when an LV-type model is appropriate. However, we recognize that these tests are not enough to predict the success of such a model—especially when handling multispecies communities. Additional research is required to determine the scope of application and the limits of such a framework.

Overall, our investigations in simple *in vitro* communities of human nasal bacteria suggest that in complex, low-nutrient environments, the quality of the environment

for the growth of each isolate can be determined by where we are along the growth rate and carrying capacity correlation line. For each isolate, points closer to the origin represent low-quality habitats for growth of that isolate and points farther away from the origin represent high-quality habitats that support fast growth rate and large carrying capacity. In this view, the growth of each isolate in the environment also modulates the quality of the habitat for other isolates, representing interbacterial interactions. These interactions can be estimated from CFSM experiments and are consistent with an LV-type model. Such a model with a central role for habitat quality is a simplification and an abstraction that does not require tracking individual interactions mediators and yet can offer some predictive power about community dynamics and outcomes (Fig. 5).

## ACKNOWLEDGMENTS

The authors would like to thank members of the K. Lemon Lab for kindly sharing their expertise with us. S.D. was supported by the NIH T32 training grant and was co-advised by Dr. Yang-Yu Liu from the Channing Division of Network Medicine at Harvard Medical School. Work in the Momeni Lab was supported by a startup fund from Boston College and by an Award for Excellence in Biomedical Research from the Smith Family Foundation. Dr. Lemon was supported by the National Institutes in Health through the National Institute of General Medical Sciences (R01 GM117174 and R35 GM141806).

S.D. and B.M. conceived the research. S.D., V.W., and B.M. designed the simulations and experiments. S.D. and V.W. ran the experiments. B.M. ran the simulations. S.D. and V.W. wrote the manuscript. K.P.L. provided expertise on human nasal bacteria. B.M. supervised the research. S.D., V.W., K.P.L., and B.M. edited the manuscript. All authors contributed to the article and approved the submitted version.

## AUTHOR AFFILIATIONS

[1]Department of Biology, Boston College, Chestnut Hill, Massachusetts, USA
[2]Department of Molecular Virology & Microbiology, Alkek Center for Metagenomics & Microbiome Research and Division of Infectious Diseases, Texas Children's Hospital, Department of Pediatrics, Baylor College of Medicine, Houston, Texas, USA

## AUTHOR ORCIDs

Katherine P. Lemon ⓘ http://orcid.org/0000-0003-1542-1679
Babak Momeni ⓘ http://orcid.org/0000-0003-1271-5196

## FUNDING

| Funder | Grant(s) | Author(s) |
| --- | --- | --- |
| HHS | NIH | National Heart, Lung, and Blood Institute (NHLBI) | 5T32HL7427-37 | Sandra Dedrick |
| Richard and Susan Smith Family Foundation (RSSFF) | Award for Excellence in Biomedical Research | Babak Momeni |
| HHS | NIH | National Institute of General Medical Sciences (NIGMS) | R01 GM117174, R35 GM141806 | Katherine P. Lemon |

## AUTHOR CONTRIBUTIONS

Sandra Dedrick, Conceptualization, Data curation, Formal analysis, Investigation, Methodology, Validation, Visualization, Writing – original draft | Vaishnavi Warrier, Data curation, Formal analysis, Investigation, Methodology, Writing – original draft, Visualization, Writing – review and editing | Katherine P. Lemon, Investigation, Methodology, Writing – review and editing, Project administration, Supervision | Babak Momeni, Conceptualization, Formal analysis, Funding acquisition, Investigation, Methodology, Project administration, Software, Supervision, Visualization, Writing – review and editing

## DATA AVAILABILITY STATEMENT

Codes related to this manuscript can be found at: https://github.com/bmomeni/nasal-community-modeling. The raw data supporting the conclusions of this article will be made available by the authors, without undue reservation.

## ADDITIONAL FILES

The following material is available online.

## Supplemental Material

**Fig S1 (286382_3_supp_6702723_rrxlxh.tif).** Growth rate and carrying capacity can be estimated by monitoring the $OD_{600}$ of cultures over time. Growth curves of *Corynebacterium* sp. KPL1821 (top) and *Staphylococcus aureus* KPL1828 (bottom) are shown as two representative strains grown at different concentrations of THY (from 2.5% to 100%). For each strain, the plot on the left (four replicates) shows the raw $OD_{600}$ values obtained from microplate reader and the plot on the right (average of four replicates) shows log-transformed values after subtracting the background (estimated as the average of the $OD_{600}$ values measured within the first 30 min). Carrying capacities are estimated based on the maximum $OD_{600}$ values within 48 h of growth (after adjustments described in Fig. S3). Growth rates are calculated by fitting a line into the log-transformed OD readings in early stages of growth (typically before $OD_{600}$ reaches 30% of its maximum value). Four replicates are used in each case.

**Fig S2 (286382_3_supp_6702724_rrxlxh.tif).** Fluorescence readings can be converted back to $OD_{600}$ in sGFP *Staphylococcus aureus* cultures. Growth rate and fluorescence are tightly linked in growing cultures of sGFP *S. aureus* Newman (left). This relation can be used to derive a calibration curve that converts fluorescence readings to corresponding cell density (using OD as a proxy), as described by the equation in Materials and Methods (right).

**Fig S3 (286382_3_supp_6702725_rrxlxh.tif).** Measured $OD_{600}$ can be used to estimate the cell density. Left: monoculture of *Staphylococcus aureus*. Right: monoculture of non-aureus *Staphylococcus* sp. KPL1850. Cells were grown to mid-exponential phase, concentrated to a high OD and then diluted back to lower ODs. Dilutions start from a measured OD of ~1.7 and cover the following dilution factors: 1, 0.5, 0.25, 0.125, 0.1, 0.08, 0.06, 0.04, 0.02, 0.01, 0.008, and 0.004. Three replicates were measured for each case. Samples diluted to an OD of 0.025 (within the linear range of OD-Density relation) were used to measure the CFUs, estimated around $1.6 \times 10^9$ cells/mL at OD 1. The conversion equation between the measure OD and the adjusted OD ($OD_{adj}$) is estimated to be $OD_{adj} = 2.7OD/(2.7\text{-}OD)$ for *S. aureus* and $OD_{adj} = 2.5OD/(2.5\text{-}OD)$ for non-aureus *Staphylococcus* sp. KPL1850. This correction is applied to adjust the OD when the measured OD was above 0.8 to more accurately estimate the interaction coefficients in cocultures. We have not adjusted the ODs in CFSM experiments (the measured $OD_{600}$ is reported in those cases), because they did not affect any of our conclusions.

**Fig S4 (286382_3_supp_6702726_rrxlxh.tif).** Growth rate-carrying capacity results from nasal isolates grown in low concentrations (5%–0.62%) of BHI show strong correlations. A linear regression analysis reveals a strong positive relationship between growth rate and carrying capacity when isolates are grown in low-nutrient concentrations. Each data point shows the average growth rate and carrying capacity (using Max $OD_{600}$ as a proxy) from 3 to 6 technical replicates.

**Fig S5 (286382_3_supp_6702727_rrxlxh.tif).** In high-nutrient environments, growth rate-carrying capacity relationships for nasal bacteria grown in different medium concentrations do not follow a simple proportionality relation. In high concentrations (100%–10%) of THY (top) and BHI (bottom) growth rate-carrying capacity relations deviate from a linear regression analysis calculated based on measured values at different concentrations. Each data point shows the average growth rate and carrying capacity (using Max $OD_{600}$ as a proxy) from 3 to 6 technical replicates.

**Fig S6 (286382_3_supp_6702728_rrxlxh.tif).** The carrying capacity of nasal isolates grown in a defined medium at various carbon concentrations is proportional to the total carbon concentration, but the growth rate is not. A linear regression analysis shows a positive correlation between carrying capacity and carbon concentrations (top). In contrast, a linear regression analysis shows only a moderate correlation between growth rate and carbon concentrations (bottom). Each data point shows the average growth rate and carrying capacity (using Max $OD_{600}$ as a proxy) from 3 to 6 technical replicates from two independent experiments. Results for *C. pseudodiphtheriticum* KPL1989 were not consistent in different experiments and are not included here.

**Fig S7 (286382_3_supp_6702729_rrxlxh.tif).** Growth rate-carrying capacity results from nasal isolates grown in the presence of lactic acid show different trends for different isolates. A linear regression analysis shows weak correlations between growth rate and carrying capacity. Each data point shows the average growth rate and carrying capacity (using Max $OD_{600}$ as a proxy) from 3 to 6 technical replicates. Different trends are observed in different isolates.

**Fig S8 (286382_3_supp_6702730_rrxlxh.tif).** Growth rate-carrying capacity results from nasal isolates grown in the presence of vancomycin show strong correlations. Each data point shows the average growth rate and carrying capacity (using Max $OD_{600}$ as a proxy) from 3 to 6 technical replicates.

**Fig S9 (286382_3_supp_6702731_rrxlxh.tif).** Comparison of experimental and modeling results for sGFP *Staphylococcus aureus* Newman (Sa) and non-aureus *Staphylococcus* sp. KPL1850 (Sna) cocultures shows that LV models fails to offer an acceptable approximation in rich media. Top: Coculture experiments and simulations were performed as described in the Materials and Methods section. Compared to 10% THY in Fig. 5, here 100% THY is used as the growth medium. Since *S. aureus* dominates these communities, initial ratios of 1:10 and 1:100 are used in this case (compared to 1:1 and 1:10 in Fig. 5). An LV model obtained from CFSM (top, dotted; bottom, dotted vs. dashed) fails to predict important trends in the dynamics. Directly fitting an LV model allows a reasonable approximation of the dynamics (top, dashed); however, the model parameters vary greatly when the initial ratio is changed, suggesting that a consistent LV model to represent the dynamics does not exist. Six technical replicates are used in each case.

**Fig S10 (286382_3_supp_6702732_rrxlxh.tif).** The impact of combined mixture of CFSMs of isolates is often similar to combined impact of each of the CFSMs. We tested sGFP *Staphylococcus aureus* (Sa), non-aureus *Staphylococcus* sp. KPL1839 (Se), and non-aureus *Staphylococcus* sp. KPL1850 (Sna) in the CFSM of each other as well as in the 1:1 combined CFSM of pairs of isolates. Our simple model predicts that the carrying capacity in the combined 1:1 CFSM (last bar in each plot) should be similar to the average of carrying capacity in each of those CFSMs (purple bar). Cases where the growth of isolates in each of the two CFSMs showed different carrying capacities are marked with an asterisk (*, *T* test). *P* values shown on each plot are for the comparison between the average 1:1 carrying capacity and the combined 1:1 carrying capacity (*T* test). Only in one out of the nine cases tested, the 1:1 carrying capacity and the combined 1:1 carrying capacity were different (Sna in Sna+Sa). In all cases, the culture medium was 10% THY. Error bars are standard deviation among three technical replicates.

## Open Peer Review

**PEER REVIEW HISTORY (review-history.pdf).** An accounting of the reviewer comments and feedback.

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
