## [Reviewer comments · mSystems]

When does a Lotka-Volterra model represent microbial interactions? Insights from *in-vitro* nasal bacterial communities

Sandra Dedrick, Vaishnavi Warriar, Katherine Lemon, and Babak Momeni

Corresponding Author(s): Babak Momeni, Boston College

Review Timeline:

Submission Date:	August 9, 2022
Editorial Decision:	October 13, 2022
Revision Received:	January 16, 2023
Editorial Decision:	February 15, 2023
Revision Received:	March 2, 2023
Editorial Decision:	March 13, 2023
Revision Received:	March 22, 2023
Accepted:	March 25, 2023

Editor: Miguel Lurgi

Reviewer(s): Disclosure of reviewer identity is with reference to reviewer comments included in decision letter(s). The following individuals involved in review of your submission have agreed to reveal their identity: Matthieu Barbier (Reviewer #2); Aurore Picot (Reviewer #3)

Transaction Report:

DOI: <https://doi.org/10.1128/msystems.00757-22>

October 13, 2022

Dr. Babak Momeni
Boston College
Biology
140 Commonwealth Ave
Higgins 346
Chestnut Hill, MA 02467

Re: mSystems00757-22 (When does a Lotka-Volterra model represent microbial interactions? Insights from *in-vitro* nasal bacterial communities)

Dear Dr. Babak Momeni:

Thank you for submitting your manuscript to mSystems. We have completed our review and I am pleased to inform you that, in principle, we expect to accept it for publication in mSystems. However, acceptance will not be final until you have adequately addressed the reviewer comments.

I found your papers very interesting. I enjoyed very much the way in which the Lotka-Volterra framework (a milestone of theoretical ecology) is tested using experimental communities.

After having myself read your manuscript and the reviewers reports I agree that some clarification is needed. In particular in terms of why it is decided that the OD600 observed are considered the carrying capacities of the systems, and also in some details and assumptions of the model. Please see the full set of comments in the reviewers reports attached and follow the instructions below when preparing your resubmission, paying particular attention to drawing a point-by-point letter demonstrating how you addressed each one of the reviewers' comments.

Preparing Revision Guidelines

Sincerely,

Miguel Lurgi

Editor, mSystems

Journals Department
Reviewer comments:

Reviewer #2 (Comments for the Author):

GENERAL ASSESSMENT

The manuscript is clear, well-written, free of errors as far as I can tell. As I am not an experimentalist, I will focus on the methodological and theoretical aspects of the work.

I think the main experimental idea, i.e. comparing growth in fresh media and cell-free spent media, is interesting and sound. The comparison to coculture experiments is what makes this work even more worthwhile in my eyes, and I really appreciate the range of information shown in Fig 5.

The way of representing and testing the adequacy of the model, i.e. looking at the relationship between initial growth and final abundance in various fresh or spent media, is nice in its simplicity, though I note below that it is not strictly a mark of the Lotka-Volterra model specifically.

Supplementary Figures covered many interesting discussion points and were all convincing (except maybe Sup Fig 11 as noted below).

I feel that I sincerely believe the authors' conclusions*, with a single true caveat:

In the current presentation, nothing really proves to me that the OD600 shown on many figures is indeed a "carrying capacity" -- i.e. I am never shown the saturation of growth outside of the coculture experiments. It could well be that I am simply looking at a transient value of abundance in the exponential growth phase, in which case a positive correlation of abundance with growth rate is rather obvious. I do not really believe this to be the case, but it would be good to lay such a doubt to rest.

To conclude, I have a few recommendations, most importantly a figure demonstrating that it is reasonable to call these OD600 "carrying capacities", at least for a reasonable fraction of them if growth curves over time are not available for all.

* As a small note concerning my own bias, I am generally sympathetic to the idea that more complex environments can lead to simpler effective models because the idiosyncrasies of any given strain-metabolite interaction are less important in aggregation. This clearly makes me more likely to believe the authors' conclusion regarding differences between simple and complex media.

MAIN COMMENTS

1) Rather than just Lotka-Volterra, any model of the form

$$dS_i / dt = r_{ij} S_j - \alpha S_i^2$$

where r_{ij} is the only factor being influenced by the fresh or spent medium, will always give proportionality between r and K , as the equilibrium abundance is

$$K_{ij} = r_{ij} / \alpha$$

There is no necessity for r_{ij} to strictly be linear in K_j .

To really test the Lotka-Volterra model, you would need to also vary K_j , i.e. do spent media experiments with various spent

media from the same species (e.g. having started with various pHs), and show that r_{ij} is indeed

$r_{ij} = r_j - c_{ij} K_j$ with constant c_{ij} .

This is not essential here, but it's worth considering that maybe a non-LV model compatible with the original observation would still work a bit better.

2) Building on this, I want to propose a slightly different way of presenting things than Lipson's explanation proposed in discussion.

First, I would like to mention that I don't think the high-nutrient results (Supp Fig 4 and 5) should be described in terms of correlations. The more striking fact is that there is little variance on the x-axis, while there is quite a lot of variance on the y-axis.

If there was little variation in both, you could have weak correlations simply due to noise and errors, but that would not say much. Here, the variance on the y-axis (and furthermore the ordering of points) mean that nutrient concentration is really doing something interesting to K.

The relationship between r and K can be clarified in the r-alpha formulation as above, i.e.

$$dS/dt = r S - \alpha S^2$$

with $K=r/\alpha$.

it seems that

-- at low nutrient concentration, alpha is roughly constant while r varies, hence $K \sim r$

-- at high nutrients, alpha really varies with nutrient concentration, while r is practically fixed.

This would be compatible while various recent studies finding strong negative per-capita interactions overall at high nutrients, e.g.

Ratzke et al (2020) Strength of species interactions determines biodiversity and stability in microbial communities
- Nature ecology & evolution

Li et al (2022) Resource availability drives bacteria community resistance to pathogen invasion via altering bacterial pairwise interactions
- Environmental Microbiology

Quoting from Ratzke: "At high nutrient concentrations, extensive microbial growth leads to strong chemical modifications of the environment, causing more negative interactions between species. "

Here this seems to be true for within-species interactions as well. (the difference with what is currently in the discussion is that there is no need to invoke a trade-off between r and K; r does not seem to be changing much anymore, except in a few instances -- instead alpha is what really is impacted)

3) the choice to re-adjust pH in the spent media is not really discussed although it means removing an important vector of species interactions, as obvious from Fig 4 and many other works (see e.g. Ratzke's paper above). I do not think it is a bad thing in itself -- indeed it might favor the expression of other mechanisms of interaction -- but it is worth noting that it is not a completely benign choice, especially as such adjustment was not done for cocultures and this may explain some of the diverging results.

More generally, thinking of interactions entirely in terms of what is left in spent media is anchored in a particular assumption about possible interaction mechanisms: it ignores any kind of interaction that would involve a more dynamical response of species' metabolisms to each other, or that is not mediated by metabolites (e.g. mechanical interactions). Thus, a quantitative disagreement with co-culture results could point to a whole range of possibilities.

4) Why not show results of Figure 2 as a matrix of interaction coefficients, instead of giving them only for the cocultured strains in Table 2? That would be an interesting result in itself for various readers (i.e. it provides an idea of the magnitude of competition or facilitation in these systems)

5) In the main text you say

"we observe a strong linear and positive correlation between carbon concentrations and carrying capacity (Supplementary Figure 6), but not growth rate (Supplementary Figure 7)"

I would not say that relationships in Sup Fig 7 are really that weak -- in most other contexts, the R squared and p-values you are showing would be taken as good results. I would simply claim the relationship to carrying capacity is stronger (and even there you had to remove an inconsistent strain), but this particular experimental manipulation is likely one where both r and K or α are changing a bit. This is not at all a main result so it is not a problem of course.

6) Regarding Sup Fig 10, I would note that Lotka-Volterra does allow a dependency on initial conditions in the case of mutual exclusion, and you would have a hard time properly fitting the coefficients and getting the right results in that case. But in any case, LV does not allow this at all when species coexist. I agree that a different model is clearly appropriate here.

7) Sup Fig 11 has honestly too little variation between media to really convince that the combined media behave like the average of the component media (rather than, say, like just one of the two component media, or like a randomly drawn spent medium)

MINOR COMMENTS

Page 2:

"To converge these distinct phenomenological and mechanistic inquiries" -> I am likely less of a proper English speaker than some authors, but this transitive use of the verb converge seems unusual.

Figure 2: Clarify in caption that colors are the media, and panels are the strain being grown (it can be gathered from the fact that 10% THY is included as a color, but still, it is better made explicit)

Reviewer #3 (Comments for the Author):

I have read the paper entitled "When does a Lotka-Volterra model represent microbial interactions? Insights from in-vitro nasal bacterial communities" with interest.

The authors use the Lotka-Volterra (LV) framework to estimate interactions from cell-free spent-media (CFSM) assays, and confront the estimated interaction with actual co-cultures experiments, using the nasal microbiota as an experimental system. The authors first derive a criteria to assess when the LV framework can be used. The growth rate and carrying capacity of monocultures (growth assays) must be linearly positively correlated in different environmental conditions.

Then, they use CFSM data to estimate pairwise interactions, and compare these predictions to cocultures. They show that LV models fitted on CFSM accurately capture the interaction when the environment is complex and limited in nutrients.

This study and the question it raises are very interesting, as the use of the LV framework to model bacterial interactions has been recently debated, in particular, in terms of its predictive ability (as referenced in the introduction). The comparison between CFSM estimations and co-cultures allows the authors to quantify this predictive power.

I do not have a lot of comments on the experimental parts (choice of medium, etc) but more on the assumptions of the modeling approach.

General comments:

1) I think there are two aspects to the study: a first one, regarding the use of LV framework to estimate interactions, and a second one, regardless of the modeling framework, regarding the use of CFSM vs cocultures. The first question is nicely introduced, and I would appreciate a bit more discussion or reference to the literature regarding the use of CFSM to estimate interactions, since it is then used as a reference to characterize the quality of the LV estimations, if I understood the approach correctly. What is the advantage of the CFSM method compared to directly fitting to the co-cultures or monocultures/co-cultures comparisons?

2) I wonder about the link between the carrying capacity as defined here in the end of introduction ("to what extent bacteria grow"), and the final biomass/yield of a Monod growth model? This comment is linked to minor comment 3. Are these two concepts equivalent?

3) A related comment on the comparison with consumer-resource models: would it be possible to estimate the interactions from fitting a Monod consumer-resource model to the monocultures and predict the co-cultures pattern with this type of data? I would be interested to see maybe in a supplementary analysis what a consumer-resource model would do in terms of fitting on top of the different curves of figure 5. Also, what framework do the authors suggest to use when LV is less relevant (i.e. when the environment is rich in nutrients or not complex)?

4) I appreciate how the authors introduce this idea of bacteria modifying their common habitat, thus inducing intra- and inter-specific interactions (beginning of page 3 & final paragraph of the discussion). This niche construction perspective reminded me of Estrela et al 2018, Environmentally Mediated Social Dilemma (<https://doi.org/10.1016/j.tree.2018.10.004>, maybe relevant to add to the reference list).

Minor comments :

1) Page 3: "This medium condition was selected since it moderately reduces the growth rate and the carrying capacity of each isolate, [...]" compared to what condition?

2) Page 8: K_i and r_i the "and" is in italics

3) Page 8: The authors state that the formulation is equivalent to the LV model in which $a_{ij} = c_{ij} r_i / K_i$. I wonder if the requirement that growth rate and carrying capacity must be linearly correlated simply comes from the fact that in the logistic formulation the carrying capacity is the ratio between an intrinsic growth and a self-regulation term, so that the two parameters are not independent. For instance, considering the alternative formulation $dS/dt = (r - a S) S$, the equilibrium density (effective carrying capacity) is $S^* = r/a$. If I use this formulation and follow the same steps as in the Supplementary Material, I find that " $r_{12}/K_{12} = a_{11}$ " which is equivalent to what the authors find and is the strength of intraspecific competition.

4) Beginning of the discussion: I think there are two "historical" LV set of equations, one for predator-prey interactions (the hare-lynx model with cycles), and one for competitive interactions with coexistence theory built upon, which was further extended to positive interactions/multispecies with generalized LV models. Maybe add "and" in "originally developed to describe competitive [and] predator-prey interaction"?

Response to Reviewers' Comments:

We thank all reviewers for their constructive suggestions. A point-by-point response to the concerns raised by the reviewers is listed in the following. In addition to the changes suggested by the reviewer, we have moved the derivation of the linear relationship between growth rate and carrying capacity values from Supplementary Information to the main text, because this was a consistent suggestion in the feedback we received from our peers.

[Authors' response is labeled as blue. Modifications to the text are shown as quoted text.]

Reviewer #2

GENERAL ASSESSMENT

The manuscript is clear, well-written, free of errors as far as I can tell. As I am not an experimentalist, I will focus on the methodological and theoretical aspects of the work.

I think the main experimental idea, i.e. comparing growth in fresh media and cell-free spent media, is interesting and sound. The comparison to coculture experiments is what makes this work even more worthwhile in my eyes, and I really appreciate the range of information shown in Fig 5.

The way of representing and testing the adequacy of the model, i.e. looking at the relationship between initial growth and final abundance in various fresh or spent media, is nice in its simplicity, though I note below that it is not strictly a mark of the Lotka-Volterra model specifically.

Supplementary Figures covered many interesting discussion points and were all convincing (except maybe Sup Fig 11 as noted below).

I feel that I sincerely believe the authors' conclusions*, with a single true caveat:

In the current presentation, nothing really proves to me that the OD600 shown on many figures is indeed a "carrying capacity" -- i.e. I am never shown the saturation of growth outside of the coculture experiments. It could well be that I am simply looking at a transient value of abundance in the exponential growth phase, in which case a positive correlation of abundance with growth rate is rather obvious. I do not really believe this to be the case, but it would be good to lay such a doubt to rest.

To conclude, I have a few recommendations, most importantly a figure demonstrating that it is reasonable to call these OD600 "carrying capacities", at least for a reasonable fraction of them if growth curves over time are not available for all.

* As a small note concerning my own bias, I am generally sympathetic to the idea that more complex environments can lead to simpler effective models because the idiosyncrasies of any given strain-metabolite interaction are less important in aggregation. This clearly makes me more likely to believe the authors' conclusion regarding differences between simple and complex media.

Thank you for your constructive feedback. You have a perfectly valid point regarding the link between OD600 and carrying capacity. We have included the growth curves (Supplementary Fig 1), along with the relationship between the OD and the actual cell numbers (Supplementary Fig 3) in the revised manuscript to give our readers the full picture. Other comments mentioned here are addressed below.

MAIN COMMENTS

1) Rather than just Lotka-Volterra, any model of the form

$$dS_i / dt = r_{ij} S_i - \alpha S_i^2$$

where r_{ij} is the only factor being influenced by the fresh or spent medium, will always give proportionality between r and K , as the equilibrium abundance is

$$K_{ij} = r_{ij} / \alpha$$

There is no necessity for r_{ij} to strictly be linear in K_j .

To really test the Lotka-Volterra model, you would need to also vary K_j , i.e. do spent media experiments with various spent media from the same species (e.g. having started with various pHs), and show that r_{ij} is indeed

$$r_{ij} = r_i - c_{ij} K_j \text{ with constant } c_{ij}.$$

This is not essential here, but it's worth considering that maybe a non-LV model compatible with the original observation would still work a bit better.

Thank you for bringing this to our attention. You are correct that there can be other models that would produce results consistent with the linear r - K relationship. As of now, our study is not ruling out that possibility. Our only definite claim is that if the linear relationship is contradicted, LV will not be the right choice. To address your point, we have added the following lines to our Discussions.

“We should emphasize that observing a linear relationship between the growth rate and carrying capacity is not sufficient to establish that a Lotka-Volterra model is appropriate for representing the community of such interactions—other models may produce the same relationship or higher-order interactions may make the multispecies LV inaccurate.

Instead we argue that if the linear relationship is not satisfied, LV will not be the right choice for representing the system."

2) Building on this, I want to propose a slightly different way of presenting things than Lipson's explanation proposed in discussion.

First, I would like to mention that I don't think the high-nutrient results (Supp Fig 4 and 5) should be described in terms of correlations. The more striking fact is that there is little variance on the x-axis, while there is quite a lot of variance on the y-axis.

If there was little variation in both, you could have weak correlations simply due to noise and errors, but that would not say much. Here, the variance on the y-axis (and furthermore the ordering of points) mean that nutrient concentration is really doing something interesting to K.

The relationship between r and K can be clarified in the r-alpha formulation as above, i.e.

$$dS/dt = r S - \alpha S^2$$

with $K=r/\alpha$.

it seems that

- at low nutrient concentration, alpha is roughly constant while r varies, hence $K \sim r$
- at high nutrients, alpha really varies with nutrient concentration, while r is practically fixed.

This would be compatible while various recent studies finding strong negative per-capita interactions overall at high nutrients, e.g.

Ratzke et al (2020) Strength of species interactions determines biodiversity and stability in microbial communities

- Nature ecology & evolution

Li et al (2022) Resource availability drives bacteria community resistance to pathogen invasion via altering bacterial pairwise interactions

- Environmental Microbiology

Quoting from Ratzke: "At high nutrient concentrations, extensive microbial growth leads to strong chemical modifications of the environment, causing more negative interactions between species. "

Here this seems to be true for within-species interactions as well. (the difference with what is currently in the discussion is that there is no need to invoke a trade-off between r and K; r does not seem to be changing much anymore, except in a few instances -- instead alpha is what really is impacted)

Thank you for this insightful suggestion. Our understanding is that you are proposing a relationship between growth rates and carrying capacities, based on alpha. Even though mathematically alpha makes perfect sense as the second order term in the equation, we could not come up with a relatable interpretation of alpha. One interpretation of alpha is the per-cell rate of resource depletion, but we are not sure if that resonates with our typical readers with a microbiology background. For that reason, we have not explicitly applied this formulation in the rest of the manuscript to avoid confusion between different parameterizations (formulations with versus without alpha).

We think overall, an interpretation based on alpha is still in-line with Lipson's interpretation, with the basic idea that at low nutrient availability, all the resources are channeled into reproduction (fixed alpha), whereas at high nutrient levels when the growth rate approaches its maximum, the cell might take alternative "more wasteful" strategies or increase its energy expenditure to deal with waste/inhibition (Ratzke 2020, Li 2022). As a result, alpha will increase.

We have also included and discussed the two papers you have mentioned for completeness and to highlight why at high nutrient availability growth rates might drop (page 20 of the 'Marked Up' manuscript, first paragraph).

"There is also additional evidence that in nutrient-rich environments populations exhibit stronger inhibitory effects (74, 75), which can lead to a drop in growth rates in cocultures compared to monocultures and thus more deviation from LV equations."

3) the choice to re-adjust pH in the spent media is not really discussed although it means removing an important vector of species interactions, as obvious from Fig 4 and many other works (see e.g. Ratzke's paper above). I do not think it is a bad thing in itself -- indeed it might favor the expression of other mechanisms of interaction -- but it is worth noting that it is not a completely benign choice, especially as such adjustment was not done for cocultures and this may explain some of the diverging results.

More generally, thinking of interactions entirely in terms of what is left in spent media is anchored in a particular assumption about possible interaction mechanisms: it ignores any kind of interaction that would involve a more dynamical response of species' metabolisms to each other, or that is not mediated by metabolites (e.g. mechanical interactions). Thus, a quantitative disagreement with co-culture results could point to a whole range of possibilities.

Regarding the pH, you are correct that a change in pH can have a significant impact on other species. In fact, in our own data we have observed that pH, if not buffered, was one of the major drivers of interactions between our strains (data not shown). However, since the environment in the nose is fairly stable (although with a distinct spatial gradient), we chose to buffer our media to remove pH as a factor. The medium in our coculture experiments is also buffered to eliminate large shifts in pH as a source of discrepancy between CFSM and coculture data. We have updated the Methods section to make this clear.

We absolutely agree with you that spent media assessment of interactions misses a range of interactions, including those that depend on physical contact, those that are strictly triggered by the partner, and mechanical interactions as you mentioned. We have added a paragraph to make this point clear and to include some of the relevant citations.

“We note that a major motivation for assessing microbial interactions using CFMS experiments is the feasibility of performing these experiments in different contexts, only requiring the ability to measure growth properties in monocultures. However, these experiments will not comprehensively capture all possible interaction mechanisms. Notable interactions such as those that rely on physical/mechanical contact between cells or those that are triggered only when a partner is present will not be captured in basic CFMS experiments. Nevertheless, because many important interactions are represented in CFMS experiments, this approach is routinely used to assess microbial interactions (17, 19, 52–54).”

4) Why not show results of Figure 2 as a matrix of interaction coefficients, instead of giving them only for the cocultured strains in Table 2? That would be an interesting result in itself for various readers (i.e. it provides an idea of the magnitude of competition or facilitation in these systems)

Thank you for the suggestion. We agree with you about the importance of such a matrix. However, we have already published such results at 10% THY elsewhere (Dedrick et al., *Frontiers in Microbiology*, 2021; <https://doi.org/10.3389/fmicb.2021.613109>), so we decided not to repeat the table in this manuscript. Instead, we included plots such as Figure 2 for a visual representation of data that is more in line with the main conclusions of this manuscript.

5) In the main text you say

"we observe a strong linear and positive correlation between carbon concentrations and carrying capacity (Supplementary Figure 6), but not growth rate (Supplementary Figure 7)"

I would not say that relationships in Sup Fig 7 are really that weak -- in most other contexts, the R squared and p-values you are showing would be taken as good results. I would simply claim the relationship to carrying capacity is stronger (and even there you had to remove an inconsistent strain), but this particular experimental manipulation is likely one where both r and K or α are changing a bit. This is not at all a main result so it is not a problem of course.

We realized this was perhaps caused by our inaccurate language. In these experiments we observed that K was linearly and proportionally related to the total carbon concentration. In contrast, r deviated from being proportional to the total carbon concentration. We have updated the captions.

6) Regarding Sup Fig 10, I would note that Lotka-Volterra does allow a dependency on initial conditions in the case of mutual exclusion, and you would have a hard time properly fitting the coefficients and getting the right results in that case. But in any case, LV does not allow this at all when species coexist. I agree that a different model is clearly appropriate here.

We agree with your statement. However, the main point of this figure was to show that we cannot find a consistent LV model to represent different initial conditions. We do not think the issue here is that the populations are lop-sided, which would be the experimental challenge in mutual exclusion cases. Here, the measured populations are large enough to offer a reasonable model in each case, but those models are inconsistent because LV is not a suitable model.

7) Sup Fig 11 has honestly too little variation between media to really convince that the combined media behave like the average of the component media (rather than, say, like just one of the two component media, or like a randomly drawn spent medium)

We agree with you that the evidence in the Supplementary Fig 11 (**Supplementary Figure 10** in the revised version) was not strong, because the differences between the two components were only significantly different in a few cases. We have de-emphasized our related conclusions (page 18 of the 'Marked Up' manuscript, the last paragraph before Discussions) in the revised version (while keeping the data).

“Although these results need further investigations, in the majority of cases tested, the response to 1:1 combined CFSMs was indistinguishable from the average of responses to each of the CFSMs (**Supplementary Figure 10**).”

MINOR COMMENTS

Page 2:

"To converge these distinct phenomenological and mechanistic inquiries" -> I am likely less of a proper English speaker than some authors, but this transitive use of the verb converge seems unusual.

Thank you. We have reworded the sentence by replacing “converge” with “reconcile” based on your suggestion.

Figure 2: Clarify in caption that colors are the media, and panels are the strain being grown (it can be gathered from the fact that 10% THY is included as a color, but still, it is better made explicit)

Thank you for the suggestion. We have now explicitly mentioned the colors and panels in the legends of Figures 2, 3, and 4.

Reviewer #3

I have read the paper entitled "When does a Lotka-Volterra model represent microbial interactions? Insights from in-vitro nasal bacterial communities" with interest.

The authors use the Lotka-Volterra (LV) framework to estimate interactions from cell-free spent-media (CFSM) assays, and confront the estimated interaction with actual co-cultures experiments, using the nasal microbiota as an experimental system.

The authors first derive a criteria to assess when the LV framework can be used. The growth rate and carrying capacity of monocultures (growth assays) must be linearly positively correlated in different environmental conditions.

Then, they use CFSM data to estimate pairwise interactions, and compare these predictions to cocultures. They show that LV models fitted on CFSM accurately capture the interaction when the environment is complex and limited in nutrients.

This study and the question it raises are very interesting, as the use of the LV framework to model bacterial interactions has been recently debated, in particular, in terms of its predictive ability (as referenced in the introduction). The comparison between CFSM estimations and co-cultures allows the authors to quantify this predictive power.

I do not have a lot of comments on the experimental parts (choice of medium, etc) but more on the assumptions of the modeling approach.

General comments:

1) I think there are two aspects to the study: a first one, regarding the use of LV framework to estimate interactions, and a second one, regardless of the modeling framework, regarding the use of CFSM vs cocultures. The first question is nicely introduced, and I would appreciate a bit more discussion or reference to the literature regarding the use of CFSM to estimate interactions, since it is then used as a reference to characterize the quality of the LV estimations, if I understood the approach correctly. What is the advantage of the CFSM method compared to directly fitting to the co-cultures or monocultures/co-cultures comparisons?

Thank you for the suggestion. We have expanded the manuscript in the revised version to mention some of the previous examples of the use of CFSM to estimate microbial interactions. This is a common approach because it is easy to implement. The only technical requirements for implementing such an assay are the ability to: (1) monitor the growth of individual species in monocultures, and (2) filter sterilize the spent media. Our choice of assay was highly driven by its simplicity, because we think this makes it the most accessible assay for a wide range of studies. We have included our motivation in the revised version.

"We note that a major motivation for assessing microbial interactions using CFSM experiments is the feasibility of performing these experiments in different contexts, only requiring the ability to measure growth properties in monocultures. However, these

experiments will not comprehensively capture all possible interaction mechanisms. Notable interactions such as those that rely on physical/mechanical contact between cells or those that are triggered only when a partner is present will not be captured in basic CFSM experiments. Nevertheless, because many important interactions are represented in CFSM experiments, this approach is routinely used to assess microbial interactions (17, 19, 52–54).”

2) I wonder about the link between the carrying capacity as defined here in the end of introduction ("to what extent bacteria grow"), and the final biomass/yield of a Monod growth model? This comment is linked to minor comment 3. Are these two concepts equivalent?

For the purpose of the discussions in this manuscript, we use the maximum OD within 48 hours of growth, adjust it using a calibration curve to account for the nonlinear relationship between OD and cell density, and use the adjusted value as an estimate of the carrying capacity. The value obtained from this approach is tightly linked to the carrying capacity defined in the basic logistic growth. The Monod growth model offers a relation between the growth rate and the concentration of a limiting resource. From the standard Monod equation, one can derive the carrying capacity as $r/a \cdot R_0$, where r is the maximum growth rate, R_0 is the initial amount of the limiting resource, and a is the amount of R_0 used to produce a cell. To the best of our knowledge, there is no direct link between the Monod coefficient (resource concentration at which growth rate reaches half of its maximum value) and the carrying capacity.

3) A related comment on the comparison with consumer-resource models: would it be possible to estimate the interactions from fitting a Monod consumer-resource model to the monocultures and predict the co-cultures pattern with this type of data? I would be interested to see maybe in a supplementary analysis what a consumer-resource model would do in terms of fitting on top of the different curves of figure 5. Also, what framework do the authors suggest to use when LV is less relevant (i.e. when the environment is rich in nutrients or not complex)?

Unfortunately, without *a priori* knowledge about the mechanisms of interactions, it is very difficult to construct a consumer-resource model based on coculture dynamics. There are many different ways to construct the model (the number of resources and their production/consumption links are the primary structural unknowns), and the measurements of population dynamics is not enough to adequately constrain the model.

4) I appreciate how the authors introduce this idea of bacteria modifying their common habitat, thus inducing intra- and inter-specific interactions (beginning of page 3 & final paragraph of the discussion). This niche construction perspective reminded me of Estrela et al 2018, Environmentally Mediated Social Dilemma (<https://doi.org/10.1016/j.tree.2018.10.004>, maybe relevant to add to the reference list).

Thank you for bringing the link to our attention. We have incorporated this in our Introduction in our revised manuscript.

Minor comments:

1) Page 3: "This medium condition was selected since it moderately reduces the growth rate and the carrying capacity of each isolate, [...]" compared to what condition?

Compared to 100% THY, where the growth rate of species is close to its maximum possible value, at 10% THY, the growth rate is large enough to be measurable, but still significantly less than the maximum possible value. As a result, in CFMSM experiments both increases and decreases in the growth rate (influenced by the CFMSM of other species) can be reliably detected. This is stated in the Methods. The level of nutrients in 10% THY is also comparable to the concentrations directly measured in the nasal passage (Krismer et al., PLOS Pathogens, 2014; <https://doi.org/10.1371/journal.ppat.1003862>). This makes this choice relevant for investigations of microbial properties *in vitro*.

2) Page 8: K_i and r_i the "and" is in italics

We have fixed this in the revised version.

3) Page 8: The authors state that the formulation is equivalent to the LV model in which $a_{ij} = c_{ij} r_i / K_i$. I wonder if the requirement that growth rate and carrying capacity must be linearly correlated simply comes from the fact that in the logistic formulation the carrying capacity is the ratio between an intrinsic growth and a self-regulation term, so that the two parameters are not independent. For instance, considering the alternative formulation $dS/dt = (r - a S) S$, the equilibrium density (effective carrying capacity) is $S^* = r/a$. If I use this formulation and follow the same steps as in the Supplementary Material, I find that " r_{12}/K_{12} " = a_{11} which is equivalent to what the authors find and is the strength of intraspecific competition.

You are correct that r/K is the magnitude of the self-regulating term in the logistic formula. The relevance to LV equations comes into picture when growth in the cell-free spent medium of other species is considered. The same ratio of r/K is obtained for growth in the cell-free spent medium obtained from other species. This allows the simplification of considering the impact as a change in "habitat quality," regardless of which species is responsible for that change.

You are correct that a linear r - K relationship is not unique to the LV model. This was also mentioned by Reviewer #2, and we have added a paragraph to the Discussion section to explicitly clarify this point.

4) Beginning of the discussion: I think there are two "historical" LV set of equations, one for predator-prey interactions (the hare-lynx model with cycles), and one for competitive interactions with coexistence theory built upon, which was further extended to positive interactions/multispecies with generalized LV models. Maybe add "and" in "originally developed to describe competitive [and] predator-prey interaction"?

Thank you for bringing this typo to our attention. It is fixed in the revised version.

February 6, 2023

Dr. Babak Momeni
Boston College
Biology
140 Commonwealth Ave
Higgins 346
Chestnut Hill, MA 02467

Re: mSystems00757-22R1 (When does a Lotka-Volterra model represent microbial interactions? Insights from *in-vitro* nasal bacterial communities)

Dear Dr. Babak Momeni:

Thank you for submitting your manuscript to mSystems. We have completed our review and I am pleased to inform you that, in principle, we expect to accept it for publication in mSystems. However, acceptance will not be final until you have adequately addressed the reviewer comments.

The reviewer comments, which you will find below, refer to 2 substantial concerns that need to be addressed before publication as they point out to misleading statements presented in the manuscript.

I would really appreciate if you could address this. I am sure you agree these changes will substantially improve the accuracy of your manuscript.

Preparing Revision Guidelines

Sincerely,

Miguel Lurgi

Editor, mSystems

Journals Department
Reviewer comments:

Reviewer #2 (Comments for the Author):

Dear Authors,

Thank you very much for your responses and updates to the manuscript!

I would suggest the following additional two minor modifications, which are simply trying to rectify statements that do not have much bearing on your main results, but are unfortunately incorrect and could be slightly misleading to your readers (not regarding the validity of your work, but more broadly).

=====

1) Most importantly, I had overlooked this previously, but you state:

"If a LV representation of interactions is valid, we would expect the growth rate-carrying capacity ratio of an isolate grown in the presence of another isolate to be equivalent to that of the isolate grown in a monoculture."

In fact that is not true in general, but only here because you are using spent media, so the focal isolate is not impacting the other one.

My recommendation: state that this proportionality of growth rate and carrying capacity with a constant slope is true here because of your experimental design using spent media, not a property of Lotka-Volterra in general.

Demonstration:

If species 1 is alone, and we write LV in the following way:

$$dN_1/dt = r_1 N_1 - A_{11} N_1^2$$

$$K_1 = r_1 / A_{11} \text{ so } r_1/K_1 = 1/A_{11}$$

If the two species are changing in abundance together:

$$dN_1/dt = r_1 N_1 - A_{11} N_1^2 - A_{12} N_2^2$$

$$dN_2/dt = r_2 N_2 - A_{21} N_1^2 - A_{22} N_2^2$$

then the initial growth rate of species 1 in presence of fully-grown species 2 ($N_2=r_2/A_{22}$) would be

$$R_1 = r_1 - A_{12} r_2/A_{22}$$

but species 1's carrying capacity in mixture would be

$$K_1 = (A_{22} r_1 - A_{12} r_2) / (A_{11} A_{22} - A_{12} A_{21})$$

so R_1/K_1 is now obviously not at all simply $1/A_{11}$ anymore, but instead $1/(A_{11} - A_{12} A_{21}/A_{22})$

(where you recover the original expression if species 2 is not impacted by species 1, i.e. $A_{21}=0$)

=====

2) This is more minor, but in your reply (and relevant parts of the main text), you state

"We think overall, an interpretation based on α is still in-line with Lipson's interpretation, with the basic idea that at low nutrient availability, all the resources are channeled into reproduction (fixed α), whereas at high nutrient levels when the growth rate approaches its maximum, the cell might take alternative "more wasteful" strategies or increase its energy expenditure to deal with waste/inhibition (Ratzke 2020, Li 2022). As a result, α will increase."

My point was that it's not clear that you actually get an r/K *tradeoff* at high nutrients (only a few of your results are compatible with such a tradeoff, and then only because of a single outlier).

Perhaps what you see is still explainable with a biological mechanism such as that suggested by Lipson, but I think it should not be framed as a tradeoff between r and K , when K is changing a lot and r not so much without a clear negative correlation between the two.

Response to Reviewers' Comments:

We thank the reviewer for pointing out the inaccuracies in the language of the manuscript. We have incorporated their suggestions accordingly, as described in the following. [Authors' response is labeled as blue. Modifications to the text are shown as quoted text.]

Reviewer #2

Dear Authors,

Thank you very much for your responses and updates to the manuscript!

I would suggest the following additional two minor modifications, which are simply trying to rectify statements that do not have much bearing on your main results, but are unfortunately incorrect and could be slightly misleading to your readers (not regarding the validity of your work, but more broadly).

=====

1) Most importantly, I had overlooked this previously, but you state:

"If a LV representation of interactions is valid, we would expect the growth rate-carrying capacity ratio of an isolate grown in the presence of another isolate to be equivalent to that of the isolate grown in a monoculture."

In fact that is not true in general, but only here because you are using spent media, so the focal isolate is not impacting the other one.

My recommendation: state that this proportionality of growth rate and carrying capacity with a constant slope is true here because of your experimental design using spent media, not a property of Lotka-Volterra in general.

Demonstration:

If species 1 is alone, and we write LV in the following way:

$$dN_1/dt = r_1 N_1 - A_{11} N_1^2$$

$$K_1 = r_1 / A_{11} \text{ so } r_1/K_1 = 1/A_{11}$$

If the two species are changing in abundance together:

$$\begin{aligned}dN_1/dt &= r_1 N_1 - A_{11} N_1^2 - A_{12} N_2^2 \\dN_2/dt &= r_2 N_2 - A_{21} N_1^2 - A_{22} N_2^2\end{aligned}$$

then the initial growth rate of species 1 in presence of fully-grown species 2 ($N_2=r_2/A_{22}$) would be

$$R_1 = r_1 - A_{12} r_2/A_{22}$$

but species 1's carrying capacity in mixture would be

$$K_1 = (A_{22} r_1 - A_{12} r_2) / (A_{11} A_{22} - A_{12} A_{21})$$

so R_1/K_1 is now obviously not at all simply $1/A_{11}$ anymore, but instead $1/(A_{11} - A_{12} A_{21}/A_{22})$

(where you recover the original expression if species 2 is not impacted by species 1, i.e. $A_{21}=0$)

Thank you for pointing this out. You are correct. The statement only holds in the supernatant assay. We have revised the sentence to make this clear (this is simply restating the equation).

"If a LV representation of interactions is valid, we would expect the growth rate-carrying capacity ratio of a focal isolate grown in the supernatant of another partner isolate to be equivalent to that of the focal isolate grown in a monoculture."

=====

2) This is more minor, but in your reply (and relevant parts of the main text), you state

"We think overall, an interpretation based on alpha is still in-line with Lipson's interpretation, with the basic idea that at low nutrient availability, all the resources are channeled into reproduction (fixed alpha), whereas at high nutrient levels when the growth rate approaches its maximum, the cell might take alternative "more wasteful" strategies or increase its energy expenditure to deal with waste/inhibition (Ratzke 2020, Li 2022). As a result, alpha will increase."

My point was that it's not clear that you actually get an r/K *tradeoff* at high nutrients (only a few of your results are compatible with such a tradeoff, and then only because of a single outlier).

Perhaps what you see is still explainable with a biological mechanism such as that suggested by Lipson, but I think it should not be framed as a tradeoff between r and K , when K is changing a lot and r not so much without a clear negative correlation between the two.

Thank you for the suggestion. We agree with you that perhaps the current statement is too strong, given the data presented in this manuscript. We have revised the relevant sentences to keep the interpretations within the scope of our observations.

“Indeed, when nasal isolates were grown in dilutions of a growth medium, we observed a positive growth rate-carrying capacity relationship at lower nutrient concentrations but not at higher nutrient concentrations. Our interpretation, consistent with Lipson’s explanation, is that in low-nutrient environments, cells dedicate all of their resources to producing biomass, leading to a strong positive correlation between growth rate and carrying capacity. In nutrient-rich environments, growth rate no longer increases linearly with the nutrient availability and as a result the growth-rate carrying capacity trend deviates from a simple linear relationship.”

March 13, 2023

Dr. Babak Momeni
Boston College
Biology
140 Commonwealth Ave
Higgins 346
Chestnut Hill, MA 02467

Re: mSystems00757-22R2 (When does a Lotka-Volterra model represent microbial interactions? Insights from *in-vitro* nasal bacterial communities)

Dear Dr. Babak Momeni:

Thank you for submitting your manuscript to mSystems. We have completed our review and I am pleased to inform you that, in principle, we expect to accept it for publication in mSystems. However, acceptance will not be final until you have adequately addressed the following issue.

I am sorry to be extra careful with this but I think it is very important for the readers to be clear on what exactly are the expectations from an LV model.

The first comment from reviewer 1 in the last revision round brings up the misleading statement of saying that r / K ratio of an isolate in co-culture is equivalent to that in isolation. To that comment you responded by changing the phrasing to this sentence:

"If a LV representation of interactions is valid, we would expect the growth rate-carrying capacity ratio of a focal isolate grown in the supernatant of another partner isolate to be equivalent to that of the focal isolate grown in a monoculture."

I don't think this new text correctly addresses the concerns of the reviewer because it still suggest the r/K equivalence. Could you please be more explicit by saying that the LV predictions do not suggest that but you only see this in this case because of the specific details of your experiment?

Also, the 'supernatant' language is hard to understand, so I recommend removing that word and explain more clearly what this means.

After the proper predictions of the LV have been laid out then you can explain why your observations depart from this predictions.

Again, I am sorry to be annoying but I think it will be much clearer that way.

Almost there!

Thanks!

Preparing Revision Guidelines

ASM policy requires that data be available to the public upon online posting of the article, so please verify all links to sequence

records, if present, and make sure that each number retrieves the full record of the data. If a new accession number is not linked or a link is broken, provide production staff with the correct URL for the record. If the accession numbers for new data are not publicly accessible before the expected online posting of the article, publication of your article may be delayed; please contact the ASM production staff immediately with the expected release date.

Sincerely,

Miguel Lurgi

Editor, mSystems

Journals Department
Reviewer comments:

Response to Reviewers' Comments:

Thank you for your comments. We have incorporated their suggestions accordingly, as described in the following.

[Authors' response is labeled as blue. Modifications to the text are shown as quoted text.]

The first comment from reviewer 1 in the last revision round brings up the misleading statement of saying that r / K ratio of an isolate in co-culture is equivalent to that in isolation. To that comment you responded by changing the phrasing to this sentence:

"If a LV representation of interactions is valid, we would expect the growth rate-carrying capacity ratio of a focal isolate grown in the supernatant of another partner isolate to be equivalent to that of the focal isolate grown in a monoculture."

I don't think this new text correctly addresses the concerns of the reviewer because it still suggests the r/K equivalence. Could you please be more explicit by saying that the LV predictions do not suggest that but you only see this in this case because of the specific details of your experiment?

Thank you for raising this concern. To be clear, our claim in the paper is that "If a LV representation of interactions is valid, we would expect the growth rate-carrying capacity ratio of a focal isolate grown in the **cell-free spent medium** of another partner isolate to be equivalent to that of the focal isolate grown in a monoculture." We do not make any assertions about the equivalency between r/K in cocultures versus monocultures.

It is our understanding (based on the accompanied derivations by the reviewer) that their concern was not aimed at r/K ratio in general. Instead they had correctly highlighted the imprecision in language ("filtrate" versus "spent medium") that gave the impression that the ratio would be consistent in *any* cell-free filtrate. Our derivations indeed show that r/K in isolation is equal to that in cell-free "spent" filtrates (i.e. in cell-free spent medium after growth of a partner to saturation). Please see the "Derivation of a linear relationship between growth rate and carrying capacity of CFSMs based on a Lotka-Volterra formulation of interactions" section on Page 6.

We realize that the confusion still existed after our previous correction, therefore we have revised the manuscript again in several locations to avoid a language that would cause this confusion.

Page 1 (Abstract), lines 13-18:

"Here, we propose that a set of simple *in vitro* experiments--growing each member in cell-free spent medium obtained from other members--can be used as a test to decide whether a LV model is appropriate for describing microbial interactions of interest. We show that for LV to be

a good candidate, the ratio of growth rate to carrying capacity of each isolate when grown in the cell-free spent media of other isolates should remain constant.”

Page 6, line 162:

“Calculating the parameters obtained from the CFMS, growth of isolate 1 in the CFMS of isolate 2 can be represented as

$$\frac{dS_1}{dt} = r_1 \left(1 - \frac{S_1 + c_{12}K_2}{K_1}\right) S_1$$

since in the CFMS of isolate 2 the environment will resemble the situation at which the density of S_2 has reached K_2 .”

Page 7, lines 177-178:

“In other words, the ratio of growth rate to carrying capacity in CFMS assays is independent of the isolate used for the CFMS. This r/K ratio is the slope of the line found in the empirical data.”

Also, the 'supernatant' language is hard to understand, so I recommend removing that word and explain more clearly what this means.

Thank you for the suggestion. We have removed the term and replaced it with CFMS, which is the more accurate term in this context. The only place that “supernatant” is used in the revised manuscript is in reference to the liquid phase after doing a centrifuge (and before filtering it).

After the proper predictions of the LV have been laid out then you can explain why your observations depart from this predictions.

Our prediction is that if the cell-free spent medium (CFMS) characterization of growth of a species in the spent medium obtained from other partners indicates a major deviation from r/K value obtained in isolation, a LV model is not a suitable model. The coculture experiments in Fig 5 and Fig S9 validate this claim: the population dynamics are consistent with a LV model in nutrient-poor environments (when r/K in CFMSs is relatively fixed) but not in nutrient-rich environments (when r/K in CFMSs is not fixed).

March 25, 2023

Dr. Babak Momeni
Boston College
Biology
140 Commonwealth Ave
Higgins 346
Chestnut Hill, MA 02467

Re: mSystems00757-22R3 (When does a Lotka-Volterra model represent microbial interactions? Insights from *in-vitro* nasal bacterial communities)

Dear Dr. Babak Momeni:

Thank you for incorporating the latest corrections to the manuscript. I think now everything is much clearer and well explained.

It was a pleasure to be able to read your work. I really enjoyed the paper.

Your manuscript has been accepted, and I am forwarding it to the ASM Journals Department for publication. For your reference, ASM Journals' address is given below. Before it can be scheduled for publication, your manuscript will be checked by the mSystems production staff to make sure that all elements meet the technical requirements for publication. They will contact you if anything needs to be revised before copyediting and production can begin. Otherwise, you will be notified when your proofs are ready to be viewed.

If you would like to submit a potential Featured Image, please email a file and a short legend to msystems@asmusa.org. Please note that we can only consider images that (i) the authors created or own and (ii) have not been previously published. By submitting, you agree that the image can be used under the same terms as the published article. File requirements: square dimensions (4" x 4"), 300 dpi resolution, RGB colorspace, TIF file format.

We recognize that the video files can become quite large, and so to avoid quality loss ASM suggests sending the video file via <https://www.wetransfer.com/>. When you have a final version of the video and the still ready to share, please send it to mSystems

staff at mSystems@asmusa.org.

Sincerely,

Miguel Lurgi
Editor, mSystems

Journals Department
E-mail: mSystems@asmusa.org